# Weighted Empirical Risk Minimization: Transfer Learning based on Importance Sampling

## Abstract

We consider statistical learning problems, when the distribution $P'$ of the training observations $Z'_1, \ldots, Z'_n$ differs from the distribution $P$ involved in the risk one seeks to minimize (referred to as the *test distribution*) but is still defined on the same measurable space as $P$ and dominates it. In the unrealistic case where the likelihood ratio $\Phi(z) = dP/dP'(z)$ is known, one may straightforwardly extends the Empirical Risk Minimization (ERM) approach to this specific *transfer learning* setup using the same idea as that behind Importance Sampling, by minimizing a weighted version of the empirical risk functional computed from the 'biased' training data $Z'_i$ with weights $\Phi(Z'_i)$. Although the *importance function* $\Phi(z)$ is generally unknown in practice, we show that, in various situations frequently encountered in practice, it takes a simple form and can be directly estimated from the $Z'_i$'s and some auxiliary information on the statistical population $P$. By means of linearization techniques, we then prove that the generalization capacity of the approach aforementioned is preserved when plugging the resulting estimates of the $\Phi(Z'_i)$'s into the weighted empirical risk. Beyond these theoretical guarantees, numerical results provide strong empirical evidence of the relevance of the approach promoted in this article.

## 1 Introduction

Prediction problems are of major importance in statistical learning. The main paradigm of predictive learning is *Empirical Risk Minimization* (ERM in abbreviated form), see *e.g.* Devroye et al. (1996). In the standard setup, $Z$ is a random variable (r.v. in short) that takes its values in a feature space $\mathcal{Z}$ with distribution $P$, $\Theta$ is a parameter space and $\ell : \Theta \times \mathcal{Z} \to \mathbb{R}_+$ is a (measurable) loss function. The risk is then defined by: $\forall \theta \in \Theta$,

$$\mathcal{R}_P(\theta) = \mathbb{E}_P \left[ \ell(\theta, Z) \right], \tag{1}$$

and more generally for any measure $Q$ on $\mathcal{Z}$: $\mathcal{R}_Q(\theta) = \int_{\mathcal{Z}} \ell(\theta, z) dQ(z)$. In most practical situations, the distribution $P$ involved in the definition of the risk is unknown and learning is based on the sole observation of an independent and identically distributed (i.i.d.) sample $Z_1, \ldots, Z_n$ drawn from $P$ and the risk (1) must be replaced by an empirical counterpart (or a possibly smoothed/penalized version of it), typically:

$$\widehat{\mathcal{R}}_P(\theta) = \frac{1}{n} \sum_{i=1}^n \ell(\theta, Z_i) = \mathcal{R}_{\widehat{P}_n}(\theta), \tag{2}$$

where $\widehat{P}_n = (1/n) \sum_{i=1}^n \delta_{Z_i}$ is the empirical measure of $P$ and $\delta_z$ denotes the Dirac measure at any point $z$. With the design of successful algorithms such as neural networks, support vector machines or boosting methods to perform ERM, the practice of predictive learning has recently received a significant attention and is now supported by a sound theory based on results in empirical process theory. The performance of minimizers of (2) can be indeed studied by means of concentration inequalities, quantifying the fluctuations of the maximal deviations $\sup_{\theta \in \Theta} |\widehat{\mathcal{R}}_P(\theta) - \mathcal{R}_P(\theta)|$ under various complexity assumptions for the functional class $\mathcal{F} = \{\ell(\theta, \cdot) : \theta \in \Theta\}$ (*e.g.* VC dimension, metric entropies, Rademacher averages), see Boucheron et al. (2013) for instance. Although, in the Big Data era, the availability of massive digitized information to train predictive rules is an undeniable opportunity for the widespread deployment of machine-learning solutions, the poor control of the data acquisition process one is confronted with in many applications puts practicioners at risk of jeopardizing the generalization ability of the rules produced by the algorithms implemented. Bias

selection issues in machine-learning are now the subject of much attention in the literature, see Bolukbasi et al. (2016), Zhao et al. (2017), Burns et al. (2019), Liu et al. (2016) or Huang et al. (2007). In the context of face analysis, a research area including a broad range of applications such as face detection, face recognition or face attribute detection, machine learning algorithms trained with baised training data, *e.g.* in terms of gender or ethnicity, raise concerns about fairness in machine learning. Unfair algorithms may induce systemic undesired disadvantages for specific social groups, see Das et al. (2018) for further details. Several examples of bias in deep learning based face recognition systems are discussed in Nagpal et al. (2019).

Throughout the present article, we consider the case where the i.i.d. sample $Z_1', \ldots, Z_n'$ available for training is not drawn from $P$ but from another distribution $P'$, with respect to which $P$ is absolutely continuous, and the goal pursued is to set theoretical grounds for the application of ideas behind Importance Sampling (IS in short) methodology to extend the ERM approach to this learning setup. We highlight that the problem under study is a very particular case of *Transfer Learning* (see *e.g.* Pan & Yang (2010), Ben-David et al. (2010) and Storkey (2009)), a research area currently receiving much attention in the literature and encompassing general situations where the information/knowledge one would like to transfer may take a form in the *target* space very different from that in the *source* space (referred to as *domain adaptation*).

**Weighted ERM (WERM).** In this paper, we investigate conditions guaranteeing that values for the parameter $\theta$ that nearly minimize (1) can be obtained through minimization of a weighted version of the empirical risk based on the $Z_i'$'s, namely

$$\widetilde{\mathcal{R}}_{w,n}(\theta) = \mathcal{R}_{\widetilde{P}_{w,n}}(\theta), \tag{3}$$

where $\widetilde{P}_{w,n} = (1/n) \sum_{i=1}^{n} w_i \delta_{Z_i'}$ and $w = (w_1, \ldots, w_n) \in \mathbb{R}_+^n$ is a certain weight vector. Of course, ideal weights $w^*$ are given by the likelihood function $\Phi(z) = (dP/dP')(z)$: $w_i^* = \Phi(Z_i')$ for $i \in \{1, \ldots, n\}$. In this case, the quantity (3) is obviously an unbiased estimate of the true risk (1):

$$\mathbb{E}_{P'}\left[ \mathcal{R}_{\widetilde{P}_{w^*,n}}(\theta) \right] = \mathcal{R}_P(\theta), \tag{4}$$

and generalization bounds for the $\mathcal{R}_P$-risk excess of minimizers of $\widetilde{\mathcal{R}}_{w^*,n}$ can be directly established by studying the concentration properties of the empirical process related to the $Z_i'$'s and the class of functions $\{\Phi(\cdot)\ell(\theta, \cdot) : \theta \in \Theta\}$ (see section 2 below). However, the *importance function* $\Phi$ is unknown in general, just like distribution $P$. It is the major purpose of this article to show that, in far from uncommon situations, the (ideal) weights $w_i^*$ can be estimated from the $Z_i'$s combined with auxiliary information on the target population $P$. As shall be seen below, such favorable cases include in particular classification problems where class probabilities in the test stage differ from those in the training step, risk minimization in stratified populations (see Bekker & Davis (2018)), with strata statistically represented in a different manner in the test and training populations, positive-unlabeled learning (PU-learning, see *e.g.* du Plessis et al. (2014)). In each of these cases, we show that the stochastic process obtained by plugging the weight estimates in the weighted empirical risk functional (3) is much more complex than a simple empirical process (*i.e.* a collection of i.i.d. averages) but can be however studied by means of *linearization techniques*, in the spirit of the ERM extensions established in Clémençon et al. (2008) or Clémençon & Vayatis (2009). Learning rate bounds for minimizers of the corresponding risk estimate are proved and, beyond these theoretical guarantees, the performance of the weighted ERM approach is supported by convincing numerical results.

The article is structured as follows. In section 2, the ideal case where the importance function $\Phi$ is known is preliminarily considered and a first basic example where the optimal weights can be easily inferred and plugged into the risk without deteriorating the learning rate is discussed. The main results of the paper are stated in section 3, which shows that the methodology promoted can be applied to two important problems in practice, risk minimization in stratified populations and PU-learning, with generalization guarantees. Illustrative numerical experiments are displayed in section 4, while some concluding remarks are collected in section 5. Proofs and additional results are deferred to the Supplementary Material.

## 2 Importance Sampling - Risk Minimization with Biased Data

Here and throughout, the indicator function of any event $\mathcal{E}$ is denoted by $\mathbb{I}\{\mathcal{E}\}$, the sup norm of any bounded function $h : \mathcal{Z} \to \mathbb{R}$ by $\|h\|_\infty$. We place ourselves in the framework of statistical learning

based on biased training data previously introduced. As a first go, we consider the unrealistic situation where the importance function $\Phi$ is known, insofar as we shall subsequently develop techniques aiming at mimicking the minimization of the ideally weighted empirical risk

$$\widetilde{\mathcal{R}}_{w^*,n}(\theta) = \frac{1}{n} \sum_{i=1}^{n} w_i^* \ell(\theta, Z_i'), \tag{5}$$

namely the (unbiased) Importance Sampling estimator of (1) based on the instrumental data $Z_1', \ldots, Z_n'$. The following result describes the performance of minimizers $\widetilde{\theta}_n^*$ of (5). Since the goal of this paper is to promote the main ideas of the approach rather than to state results with the highest level of generality due to space limitations, we assume throughout the article for simplicity that $\ell$ and $\Phi$ are both bounded functions. For $\sigma_1, \ldots, \sigma_n$ independent Rademacher random variables (*i.e.* symmetric $\{-1, 1\}$-valued r.v.'s), independent from the $Z_i'$'s, we define the Rademacher average associated to the class of function $\mathcal{F}$ as $R_n'(\mathcal{F}) := \mathbb{E}_\sigma \left[ \sup_{\theta \in \Theta} \frac{1}{n} \left| \sum_{i=1}^{n} \sigma_i \ell(\theta, Z_i') \right| \right]$. This quantity can be bounded by metric entropy methods under appropriate complexity assumptions on the class $\mathcal{F}$, it is for instance of order $O_\mathbb{P}(1/\sqrt{n})$ when $\mathcal{F}$ is a VC major class with finite VC dimension, see *e.g.* Boucheron et al. (2005).

**Lemma 1.** *With probability at least* $1 - \delta$, *we have:* $\forall n \geq 1$,

$$\mathcal{R}_P(\widetilde{\theta}_n^*) - \min_{\theta \in \Theta} \mathcal{R}_P(\theta) \leq 4\|\Phi\|_\infty \mathbb{E}\left[R_n'(\mathcal{F})\right] + 2\|\Phi\|_\infty \sup_{(\theta,z) \in \Theta \times \mathcal{Z}} \ell(\theta, z) \sqrt{\frac{2\log(1/\delta)}{n}}.$$

Of course, when $P' = P$, we have $\Phi \equiv 1$ and the bound stated above simply describes the performance of standard empirical risk minimizers. The proof is based on the standard bound

$$\mathcal{R}_P(\widetilde{\theta}_n^*) - \min_{\theta \in \Theta} \mathcal{R}_P(\theta) \leq 2 \sup_{\theta \in \Theta} \left| \widetilde{\mathcal{R}}_{w^*,n}(\theta) - \mathbb{E}\left[\widetilde{\mathcal{R}}_{w^*,n}(\theta)\right] \right|,$$

combined with basic concentration results for empirical processes, see the Supplementary Material for further details. Of course, the importance function $\Phi$ is generally unknown and must be estimated in practice. As illustrated by the elementary example below (related to binary classification, in the situation where the probability of occurence of a positive instance significantly differs in the training and test stages), in certain statistical learning problems with biased training distribution, $\Phi$ takes a simplistic form and can be easily estimated from the $Z_i'$'s combined with auxiliary information on $P$.

**Binary classification with varying class probabilities.** The flagship problem in supervised learning corresponds to the simplest situation, where $Z = (X, Y)$, $Y$ being a binary variable valued in $\{-1, +1\}$ say, and the r.v. $X$ takes its values in a measurable space $\mathcal{X}$ and models some information hopefully useful to predict $Y$. The parameter space $\Theta$ is a set $\mathcal{G}$ of measurable mappings (*i.e.* classifiers) $g : \mathcal{X} \to \{-1, +1\}$ and the loss function is given by $\ell(g, (x,y)) = \mathbb{I}\{g(x) \neq y\}$ for all $g$ in $\mathcal{G}$ and any $(x, y) \in \mathcal{X} \times \{-1, +1\}$. The distribution $P$ of the random pair $(X, Y)$ can be either described by $X$'s marginal distribution $\mu(dx)$ and the posterior probability $\eta(x) = \mathbb{P}\{Y = +1 \mid X = x\}$ or else by the triplet $(p, F_+, F_-)$ where $p = \mathbb{P}\{Y = +1\}$ and $F_\sigma(dx)$ is $X$'s conditional distribution given $Y = \sigma 1$ with $\sigma \in \{-, +\}$. It is very common that the fraction of positive instances in the training dataset is significantly lower than the rate $p$ expected in the test stage, supposed to be known here (see the Supplementary Material for the case where the rate $p$ is only approximately known). We thus consider the case where the distribution $P'$ of the training data $(X_1', Y_1'), \ldots, (X_n', Y_n')$ is described by the triplet $(p', F_+, F_-)$ with $p' < p$. The likelihood function takes the simple following form

$$\Phi(x, y) = \mathbb{I}\{y = +1\}\frac{p}{p'} + \mathbb{I}\{y = -1\}\frac{1-p}{1-p'} \overset{def}{=} \phi(y),$$

which reveals that it depends on the label $y$ solely, and the ideally weighted empirical risk process is

$$\widetilde{\mathcal{R}}_{w^*,n}(g) = \frac{p}{p'}\frac{1}{n} \sum_{i:Y_i'=1} \mathbb{I}\{g(X_i') = -1\} + \frac{1-p}{1-p'}\frac{1}{n} \sum_{i:Y_i'=-1} \mathbb{I}\{g(X_i') = +1\}. \tag{6}$$

In general the theoretical rate $p'$ is unknown and one replaces (6) with

$$\widetilde{\mathcal{R}}_{\widehat{w}^*,n}(g) = \frac{p}{n_+'} \sum_{i:Y_i'=1} \mathbb{I}\{g(X_i') = -1\} + \frac{1-p}{n_-'} \sum_{i:Y_i'=-1} \mathbb{I}\{g(X_i') = +1\}, \tag{7}$$

where $n'_+ = \sum_{i=1}^n \mathbb{I}\{Y'_i = +1\} = n - n'_-$, $\widehat{w}_i^* = \widehat{\phi}(Y'_i)$ and $\widehat{\phi}(y) = \mathbb{I}\{y = +1\}np/n'_+ + \mathbb{I}\{y = -1\}n(1-p)/n'_-$. The stochastic process above is not a standard empirical process but a collection of sums of two ratios of basic averages. However, the following result provides a uniform control of the deviations between the ideally weighted empirical risk and that obtained by plugging the empirical weights into the latter.

**Lemma 2.** *Let $\varepsilon \in (0, 1/2)$. Suppose that $p' \in (\varepsilon, 1-\varepsilon)$. For any $\delta \in (0,1)$, we have with probability larger than $1 - \delta$:*

$$\sup_{g \in \mathcal{G}} \left| \widetilde{\mathcal{R}}_{\widehat{w}^*,n}(g) - \widetilde{\mathcal{R}}_{w^*,n}(g) \right| \leq \frac{2}{\varepsilon^2} \sqrt{\frac{\log(2/\delta)}{2n}},$$

*as soon as $n \geq 2\log(2/\delta)/\varepsilon^2$.*

See the Appendix for the technical proof. Consequently, minimizing (7) nearly boils down to minimizing (6). Combining Lemmas 2 and 1, we immediately get the generalization bound stated in the result below.

**Corollary 1.** *Suppose that the hypotheses of Lemma 2 are fulfilled. Let $\widetilde{g}_n$ be any minimizer of $\widetilde{\mathcal{R}}_{\widehat{w}^*,n}$ over class $\mathcal{G}$. We have with probability at least $1 - \delta$:*

$$\mathcal{R}_P(\widetilde{g}_n) - \inf_{g \in \mathcal{G}} \mathcal{R}_P(g) \leq \frac{2\max(p, 1-p)}{\varepsilon} \left( 2\mathbb{E}[R'_n(\mathcal{G})] + \sqrt{\frac{2\log(2/\delta)}{n}} \right) + \frac{4}{\varepsilon^2} \sqrt{\frac{\log(4/\delta)}{2n}},$$

*as soon as $n \geq 2\log(4/\delta)/\varepsilon^2$; where $R'_n(\mathcal{G}) = (1/n)\mathbb{E}_\sigma[\sup_{g \in \mathcal{G}} | \sum_{i=1}^n \sigma_i \mathbb{I}\{g(X'_i) \neq Y'_i\}|]$.*

Hence, some side information (*i.e.* knowledge of parameter $p$) has permitted to weight the training data in order to build an empirical risk functional that approximates the target risk and to show that minimization of this risk estimate yields prediction rules with optimal (in the minimax sense) learning rates. The purpose of the subsequent analysis is to show that this remains true for more general problems. Observe in addition that the bound in Corollary 1 deteriorates as $\varepsilon$ decays to zero: the method used here is not intended to solve the *few shot* learning problem, where almost no training data with positive labels is available (*i.e.* $p' \approx 0$). As shall be seen in subsection 3.2, alternative estimators of the importance function must be considered in this situation.

**Remark 1.** *Although the quantity (7) can be viewed as a cost-sensitive version of the empirical classification risk based on the $(X'_i, Y'_i)$'s (see e.g. Bach et al. (2006)), we point out that the goal pursued here is not to achieve an appropriate trade-off between type I and type II errors in the $P'$ classification problem as in biometric applications for instance (i.e. optimization of the $(F_+, F_-)$-ROC curve at a specific point) but to transfer knowledge gained in analyzing the biased data drawn from $P'$ to the classification problem related to distribution $P$.*

**Related work.** We point out that the natural idea of using weights in ERM problems that mimic those induced by the importance function has already been used in Sugiyama et al. (2008) for *covariate shift adaptation* problems (*i.e.* supervised situations, where the conditional distribution of the output given the input information is the same in the training and test domains), when, in contrast to the framework considered here, a test sample is additionally available (a method for estimating directly the importance function based on Kullback-Leibler divergence minimization is proposed, avoiding estimation of the test density). Importance sampling estimators have been also considered in Garcke & Vanck (2014) in the setup of *inductive transfer learning* (the tasks between source and target are different, regardless of the similarities between source and target domains), where the authors have proposed two methods to approximate the importance function, among which one is again based on minimizing the Kullback-Leibler divergence between the two distributions. In Cortes et al. (2008), the sample selection bias is assumed to be independent from the label, which is not true under our stratum-shift assumption or for the PU learning problem (see section 3). Lemma 1 assumes that the exact importance function is known, as does Cortes et al. (2010). The next section introduces new results for more realistic settings where it has to be learned from the data.

## 3 WEIGHTED EMPIRICAL RISK MINIMIZATION - GENERALIZATION GUARANTEES

Through two important and generic examples, relevant for many applications, we show that the approach sketched above can be applied to general situations, where appropriate auxiliary information on the target distribution is available, with generalization guarantees.

### 3.1 Statistical Learning from Biased Data in a Stratified Population

A natural extension of the simplistic problem considered in section 2 is multiclass classification in a stratified population. The random labels $Y$ and $Y'$ are supposed to take their values in $\{1, \ldots, J\}$ say, with $J \geq 1$, and each labeled observation $(X, Y)$ belongs to a certain random stratum $S$ in $\{1, \ldots, K\}$ with $K \geq 1$. Again, the distribution $P$ of a random element $Z = (X, Y, S)$ may be described by the parameters $\{(p_{j,k}, F_{j,k}) : 1 \leq j \leq J, 1 \leq k \leq K\}$ where $F_{j,k}$ is the conditional distribution of $X$ given $(Y, S) = (j, k)$ and $p_{j,k} = \mathbb{P}_{(X,Y,S) \sim P}\{Y = j, S = k\}$. Then, we have

$$dP(x, y, s) = \sum_{j=1}^{J} \sum_{k=1}^{K} \mathbb{I}\{y = j, s = k\} p_{j,k} dF_{j,k}(x),$$

and considering a distribution $P'$ with $F_{j,k} \equiv F'_{j,k}$ but possibly different class-stratum probabilities $p'_{j,k}$, the likelihood function becomes

$$\frac{dP}{dP'}(x, y, s) = \sum_{j=1}^{J} \sum_{k=1}^{K} \frac{p_{j,k}}{p'_{j,k}} \mathbb{I}\{y = j, s = k\} \overset{def}{=} \phi(y, s).$$

A more general framework can actually encompass this specific setup by defining 'meta-strata' in $\{1, \ldots, J\} \times \{1, \ldots, K\}$. Strata may often correspond to categorical input features in practice. The formalism introduced below is more general and includes the example considered in the preceding section, where strata are defined by labels.

**Learning from biased stratified data.** Consider a general mixture model, where distributions $P$ and $P'$ are stratified over $K \geq 1$ strata. Namely, $Z = (X, S)$ and $Z' = (X', S')$ with auxiliary random variables $S$ and $S'$ (the strata) valued in $\{1, \ldots, K\}$. We place ourselves in a *stratum-shift* context, assuming that the conditional distribution of $X$ given $S = k$ is the same as that of $X'$ given $S' = k$, denoted by $F_k(dx)$, for any $k \in \{1, \ldots, K\}$. However, stratum probabilities $p_k = \mathbb{P}(S = k)$ and $p'_k = \mathbb{P}(S' = k)$ may possibly be different. In this setup, the likelihood function depends only on the strata and can be expressed in a very simple form, as follows:

$$\frac{dP}{dP'}(x, s) = \sum_{k=1}^{K} \mathbb{I}\{s = k\} \frac{p_k}{p'_k} \overset{def}{=} \phi(s).$$

In this case, the ideally weighted empirical risk writes

$$\widetilde{\mathcal{R}}_{w^*, n}(\theta) = \frac{1}{n} \sum_{i=1}^{n} \ell(\theta, Z'_i) \sum_{k=1}^{K} \mathbb{I}\{S'_i = k\} \frac{p_k}{p'_k}.$$

If the strata probabilities $p_k$'s for the test distribution are known, an empirical counterpart of the ideal empirical risk above is obtained by simply plugging estimates of the $p'_k$'s computed from the training data:

$$\widetilde{\mathcal{R}}_{\widehat{w}^*, n}(\theta) = \sum_{i=1}^{n} \ell(\theta, Z'_i) \sum_{k=1}^{K} \mathbb{I}\{S'_i = k\} \frac{p_k}{n'_k}, \tag{8}$$

with $n'_k = \sum_{i=1}^{n} \mathbb{I}\{S'_i = k\}$, $\widehat{w}^*_i = \widehat{\phi}(S'_i)$ and $\widehat{\phi}(s) = \sum_{k=1}^{K} \mathbb{I}\{s = k\} n p_k / n'_k$.

A bound for the excess of risk is given in Theorem 1, that can be viewed as a generalization of Corollary 1.

**Theorem 1.** *Let $\varepsilon \in (0, 1/2)$ and assume that $p'_k \in (\varepsilon, 1 - \varepsilon)$ for $k = 1, \ldots, K$. Let $\widetilde{\theta}^*_n$ be any minimizer of $\widetilde{\mathcal{R}}_{\widehat{w}^*, n}$ as defined in (8) over class $\Theta$. We have with probability at least $1 - \delta$:*

$$\mathcal{R}_P(\widetilde{\theta}^*_n) - \inf_{\theta \in \Theta} \mathcal{R}_P(\theta) \leq \frac{2 \max_k p_k}{\varepsilon} \left( 2\mathbb{E}[R'_n(\mathcal{F})] + L \sqrt{\frac{2 \log(2/\delta)}{n}} \right) + \frac{4L}{\varepsilon^2} \sqrt{\frac{\log(4K/\delta)}{2n}},$$

*as soon as $n \geq 2 \log(4K/\delta)/\epsilon^2$; where $R'_n(\mathcal{F}) = (1/n)\mathbb{E}_\sigma[\sup_{\theta \in \Theta} | \sum_{i=1}^{n} \sigma_i \ell(\theta, Z'_i)|]$, and the loss is bounded by $L = \sup_{(\theta, z) \in \Theta \times \mathcal{Z}} \ell(\theta, z)$.*

Just like in Corollary 1, the bound in Theorem 1 explodes when $\varepsilon$ vanishes, which corresponds to the situation where a stratum $k \in \{1, \ldots, K\}$ is very poorly represented in the training data, *i.e.* when $p'_k \ll p_k$. Again, as highlighted by the experiments carried out, reweighting the losses in a frequentist (ERM) approach guarantees good generalization properties in a specific setup only, where the training information, though biased, is sufficiently informative.

### 3.2 Positive-Unlabeled Learning

Relaxing the *stratum-shift* assumption made in the previous subsection, the importance function becomes more complex and writes:

$$\Phi(x, s) = \frac{dP}{dP'}(x, s) = \sum_{k=1}^{K} \mathbb{I}\{s = k\} \frac{p_k}{p'_k} \frac{dF_k}{dF'_k}(x),$$

where $F_k$ and $F'_k$ are respectively the conditional distributions of $X$ given $S = k$ and of $X'$ given $S' = k$. The Positive-Unlabeled (PU) learning problem, which has recently been the subject of much attention (see *e.g.* du Plessis et al. (2014), Du Plessis et al. (2015), Kiryo et al. (2017)), provides a typical example of this situation. Re-using the notations introduced in section 2, in the PU problem, the testing and training distributions $P$ and $P'$ are respectively described by the triplets $(p, F_+, F_-)$ and $(q, F_+, F)$, where $F = pF_+ + (1 - p)F_-$ is the marginal distribution of $X$. Hence, the objective pursued is to solve a binary classification task, based on the sole observation of a training sample pooling data with positive labels and unlabeled data, $q$ denoting the theoretical fraction of positive data among the dataset. As noticed in du Plessis et al. (2014) (see also Du Plessis et al. (2015), Kiryo et al. (2017)), the likelihood/importance function can be expressed in a simple manner, as follows:

$$\forall (x, y) \in \mathcal{X} \times \{-1, +1\}, \quad \Phi(x, y) = \frac{p}{q} \mathbb{I}\{y = +1\} + \frac{1}{1 - q} \mathbb{I}\{y = -1\} - \frac{p}{1 - q} \frac{dF_+}{dF}(x) \mathbb{I}\{y = -1\}. \quad (9)$$

Based on an i.i.d. sample $(X'_1, Y'_1)$, $\ldots$, $(X'_n, Y'_n)$ drawn from $P'$ combined with the knowledge of $p$ (which can also be estimated from PU data, see e.g. Du Plessis & Sugiyama (2014)) and using that $F_- = (1/(1 - p))(F - pF_+)$, one may obtain estimators of $q$, $F_+$ and $F$ by computing $n'_+/n = (1/n) \sum_{i=1}^{n} \mathbb{I}\{Y'_i = +1\}$, $\widehat{F}_+ = (1/n'_+) \sum_{i=1}^{n} \mathbb{I}\{Y'_i = +1\} \delta_{X'_i}$ and $\widehat{F} = (1/n'_-) \sum_{i=1}^{n} \mathbb{I}\{Y'_i = -1\} \delta_{X'_i}$. However, plugging these quantities into (9) do not permit to get a statistical version of the importance function, insofar as the probability measures $\widehat{F}_+$ and $\widehat{F}$ are mutually singular with probability one, as soon as $F_+$ is continuous. Of course, as proposed in du Plessis et al. (2014), one may use statistical methods (*e.g.* kernel smoothing) to build distribution estimators, that ensures absolute continuity but are subject to the curse of dimensionality. However, WERM can still be applied in this case, by observing that: $\forall g \in \mathcal{G}$,

$$\mathcal{R}_P(g) = -p + \mathbb{E}_{P'} \left[ \frac{2p}{q} \mathbb{I}\{g(X') = -1, \ Y' = +1\} + \frac{1}{1 - q} \mathbb{I}\{g(X') = +1, \ Y' = -1\} \right], \quad (10)$$

which leads to the weighted empirical risk

$$\frac{2p}{n'_+} \sum_{i:Y'_i=+1} \mathbb{I}\{g(X'_i) = -1\} + \frac{1}{n'_-} \sum_{i:Y'_i=-1} \mathbb{I}\{g(X'_i) = +1\}. \quad (11)$$

Minimization of (11) yields rules $\overline{g}_n$ whose generalization ability regarding the binary problem related to $(p, F_+, F_-)$ can be guaranteed, as shown by the following result, the form of the weighted empirical risk in this case being quite similar to (7).

**Theorem 2.** *Let $\varepsilon \in (0, \ 1/2)$. Suppose that $q \in (\varepsilon, \ 1 - \varepsilon)$. Let $\overline{g}_n$ be any minimizer of the weighted empirical risk* (11) *over class $\mathcal{G}$. We have with probability at least $1 - \delta$:*

$$\mathcal{R}_P(\overline{g}_n) - \inf_{g \in \mathcal{G}} \mathcal{R}_P(g) \leq \frac{2 \max(2p, 1)}{\varepsilon} \left( 2\mathbb{E}[R'_n(\mathcal{G})] + \sqrt{\frac{2 \log(2/\delta)}{n}} \right) + \frac{4(2p + 1)}{\varepsilon^2} \sqrt{\frac{\log(4/\delta)}{2n}},$$

*as soon as $n \geq 2 \log(4/\delta)/\varepsilon^2$; where $R'_n(\mathcal{G}) = (1/n)\mathbb{E}_\sigma[\sup_{g \in \mathcal{G}} |\sum_{i=1}^{n} \sigma_i \mathbb{I}\{g(X'_i) \neq Y'_i\}|]$.*

**Remark 2.** *Let $\eta(x) = \mathbb{P}\{Y = +1 \mid X = x\}$ denote the posterior probability and recall that $(dF_+/dF_-)(x) = ((1 - p)/p)(\eta(x)/(1 - \eta(x)))$. Observing that*

$$\Phi(x, y) = \frac{p}{q} \mathbb{I}\{y = +1\} + \frac{1 - \eta(x)}{1 - q} \mathbb{I}\{y = -1\}, \quad (12)$$

*in the case when an estimate $\widehat{\eta}(x)$ of $\eta(x)$ is available, one can perform WERM using the empirical weight function*

$$\widehat{\Phi}(x, y) = \frac{np}{n'_+} \mathbb{I}\{y = +1\} + \frac{1 - \widehat{\eta}(x)}{1 - n'_+/n} \mathbb{I}\{y = -1\}. \quad (13)$$

*A bound that describes how this approach generalizes, depending on the accuracy of estimate $\widehat{\eta}$, can be easily established, for more details refer to Theorem 3 in the Supplementary Material, where it is also discussed how to exploit such formulas in order to design incremental WERM procedures.*

## 4 NUMERICAL EXPERIMENTS

This section illustrates the impact of reweighting by the likelihood ratio on classification performances, as a special case of the general strategy presented in Section 2. A first simple illustration on known probability distributions highlights the impact of the shapes of the distributions on the importance of reweighting. This example illustrates in the infinite-sample case that separable or almost separable data do not require reweighting, in contrast to noisy data. Since the distribution shapes are unknown for real data, we infer that reweighting will have variable effectiveness, depending on the dataset. This illustration is deferred to the Appendix, as well as an experiment on reweighting for classification of MNIST dataset where bias is introduced in the distribution of the classes. We detail here a second experiment that uses the structure of ImageNet to illustrate reweighting with a stratified population and strata distribution bias or *strata bias*. The code of the experiments can be found at https://drive.google.com/drive/folders/1-tWJ4n4WyXuTza8dLPngyHSVprKUZFVJ?usp=sharing.

We focus on the *learning from biased stratified data* setting introduced in Section 3.1 by leveraging the ImageNet Large Scale Visual Recognition Challenge (ILSVRC); a well-known benchmark for the image classification task, see Russakovsky et al. (2014) for more details.

The challenge consists in learning a classifier from 1.3 million training images spread out over 1,000 classes. Performance is evaluated using the validation dataset of 50,000 images of ILSVRC as our test dataset. ImageNet is an image database organized according to the WordNet hierarchy, which groups nouns in sets of related words called synsets. In that context, images are examples of very precise nouns, e.g. *flamingo*, which are contained in a larger synset, e.g. *bird*.

The impact of reweighting in presence of strata bias is illustrated on the ILSVRC classification problem with broad significance synsets for strata. To do this, we encode the data using deep neural networks. Specifically our encoding is the flattened output of the last convolutional layer of the network ResNet50 introduced in He et al. (2015). It was trained for classification on the training dataset of ILSVRC. The encodings $X_1, \ldots, X_n$ belong to a 2,048-dimensional space.

A total of 33 strata are derived from a list of high-level categories provided by ImageNet[1]. The construction of the strata is postponed to the Appendix. By default, strata probabilities $p_k$ and $p'_k$ for $1 \le k \le K$ are equivalent between training and testing datasets, meaning that reweighting by $\Phi$ would have little to no effect. Since our testing data is the validation data of ILSVRC, we have around 25 times more training than testing data. Introducing a strata bias parameter $0 \le \gamma \le 1$, we set the strata train probabilities such that $p'_k = \gamma^{1-\lfloor K/2 \rfloor/k} p_k$ before renormalization and remove train instances so that the train set has the right distribution over strata; see the Appendix for more details on the generation of strata bias. When $\gamma$ is close to one, there is little to no strata bias. In contrast, when $\gamma$ approaches 0, strata bias is extreme.

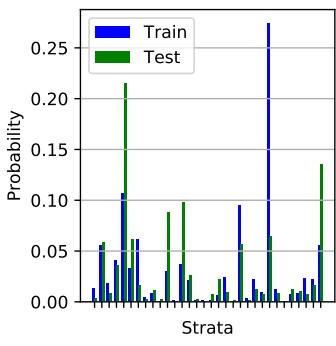

Comparison of $p_k$'s and $p'_k$'s.

| Model | Reweighting | miss rate | top-5 error |
|-------|-------------|-----------|-------------|
| Linear | Unif. $\widehat{\Phi} = 1$ | 0.344 | 0.130 |
| | Strata $\widehat{\Phi}$ | **0.329** | **0.120** |
| | Class $\widehat{\Phi}$ | 0.328 | 0.119 |
| | No bias | 0.297 | 0.102 |
| MLP | Unif. $\widehat{\Phi} = 1$ | 0.371 | 0.143 |
| | Strata $\widehat{\Phi}$ | **0.364** | **0.138** |
| | Class $\widehat{\Phi}$ | 0.363 | 0.138 |
| | No bias | 0.316 | 0.111 |

Table of results.

Figure 1: Results for the strata reweighting experiment with ImageNet.

---

[1] http://www.image-net.org/about-stats

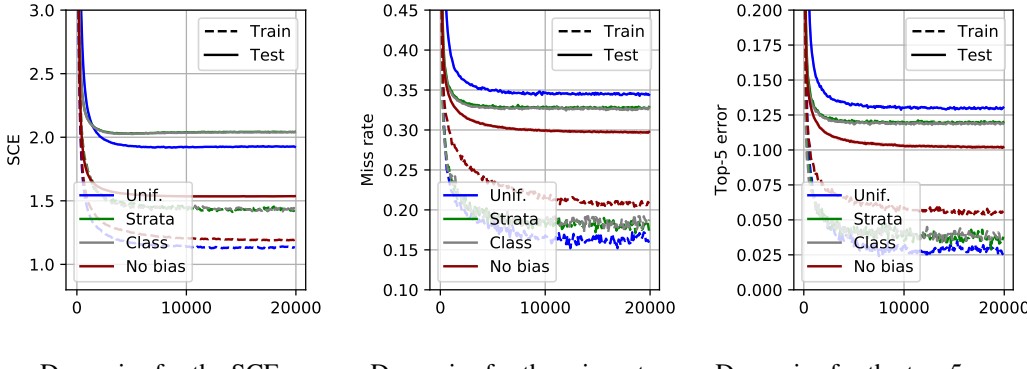

Dynamics for the SCE.          Dynamics for the miss rate.          Dynamics for the top-5 error.

Figure 2: Dynamics for the linear model for the strata reweighting experiment with ImageNet.

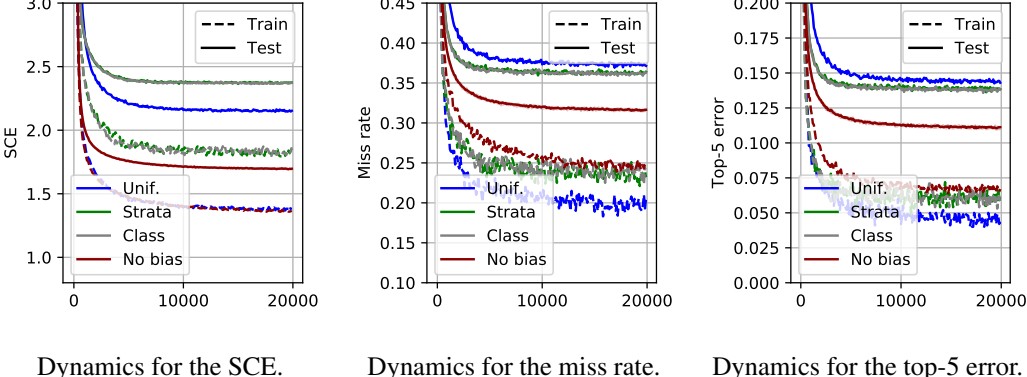

Dynamics for the SCE.          Dynamics for the miss rate.          Dynamics for the top-5 error.

Figure 3: Dynamics for the MLP model for the strata reweighting experiment with ImageNet.

The models used are a linear model and a multilayer perceptron (MLP) with one hidden layer; more details are given in the Appendix. We report better performance when reweighting using the strata information, compared to the case where the strata information is ignored, see fig. 1. For comparison, we added two reference experiments: one which reweights the train instances by the class probabilities, which we do not know in a stratified population experiment, and one with more data and no strata bias because it uses all of the ILSVRC train data. The dominance of the linear model over the MLP can be justified by the much higher number of parameters to estimate for the MLP.

## 5    CONCLUSION

In this paper, we have considered specific transfer learning problems, where the distribution of the test data $P$ differs from that of the training data, $P'$, and is absolutely continuous with respect to the latter. This setup encompasses many situations in practice, where the data acquisition process is not perfectly controlled. In this situation, a simple change of measure shows that the target risk may be viewed as the expectation of a weighted version of the basic empirical risk, with ideal weights given by the importance function $\Phi = dP/dP'$, unknown in practice. Throughout this article, we have shown that, in statistical learning problems corresponding to a wide variety of practical applications, these ideal weights can be replaced by statistical versions based solely on the training data combined with very simple information about the target distribution. The generalisation capacity of rules learnt from biased training data by minimization of the weighted empirical risk has been established, with learning bounds. These theoretical results are also illustrated with several numerical experiments.

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

## APPENDIX - TECHNICAL PROOFS

Here we detail the proofs of the results stated in the article and discuss their connection with related work.

### PROOF OF LEMMA 1

Let $\delta \in (0, 1)$. Applying the classic maximal deviation bound stated in Theorem 3.2 of Boucheron et al. (2005) to the bounded class $\mathcal{K} = \{z \in \mathcal{Z} \mapsto \Phi(z)l(\theta, z) : \quad \theta \in \Theta\}$, we obtain that, with probability at least $1 - \delta$:

$$\sup_{\theta \in \Theta} \left| \widetilde{\mathcal{R}}_{w^*,n}(\theta) - \mathbb{E}\left[\widetilde{\mathcal{R}}_{w^*,n}(\theta)\right] \right| \leq 2\mathbb{E}\left[R'_n(\mathcal{K})\right] + \|\Phi\|_\infty \sup_{(\theta,z) \in \Theta \times \mathcal{Z}} |\ell(\theta, z)| \sqrt{\frac{2\log(1/\delta)}{n}}.$$

In addition, by virtue of the contraction principle, we have $R'_n(\mathcal{K}) \leq \|\Phi\|_\infty R'_n(\mathcal{F})$ almost-surely. The desired result can be thus deduced from the bound above combined with the classic bound

$$\mathcal{R}_P(\tilde{\theta}^*_n) - \min_{\theta \in \Theta} \mathcal{R}_P(\theta) \leq 2 \sup_{\theta \in \Theta} \left| \widetilde{\mathcal{R}}_{w^*,n}(\theta) - \mathbb{E}\left[\widetilde{\mathcal{R}}_{w^*,n}(\theta)\right] \right|.$$

### PROOF OF LEMMA 2

Apply twice the Taylor expansion

$$\frac{1}{x} = \frac{1}{a} - \frac{x - a}{a^2} + \frac{(x - a)^2}{xa^2},$$

so as to get

$$
\begin{aligned}
\frac{1}{n'_+/n} &= \frac{1}{p'} - \frac{n'_+/n - p'}{p'^2} + \frac{(n'_+/n - p')^2}{p'^2 n'_+/n}, \\
\frac{1}{n'_-/n} &= \frac{1}{1-p'} - \frac{n'_-/n - 1 + p'}{(1-p')^2} + \frac{(n'_-/n - 1 + p')^2}{(1-p')^2 n'_-/n}.
\end{aligned}
$$

This yields the decomposition

$$
\widetilde{\mathcal{R}}_{\widehat{w}^*,n}(g) - \widetilde{\mathcal{R}}_{w^*,n}(g) = -\frac{p}{p'^2}\left(\frac{n'_+}{n} - p'\right)\frac{1}{n}\sum_{i=1}^{n}\mathbb{I}\{g(X'_i) = -1,\ Y'_i = +1\}
$$

$$
-\frac{1-p}{(1-p')^2}\left(\frac{n'_-}{n} - 1 + p'\right)\frac{1}{n}\sum_{i=1}^{n}\mathbb{I}\{g(X'_i) = +1,\ Y'_i = -1\} + \frac{p(n'_+/n - p')^2}{p'^2 n'_+/n}\frac{1}{n}\sum_{i=1}^{n}\mathbb{I}\{g(X'_i) = -1,\ Y'_i = +1\}
$$

$$
+ \frac{(1-p)(n'_-/n - 1 + p')^2}{(1-p')^2 n'_-/n}\frac{1}{n}\sum_{i=1}^{n}\mathbb{I}\{g(X'_i) = +1,\ Y'_i = -1\}.
$$

We deduce that

$$
\left|\widetilde{\mathcal{R}}_{\widehat{w}^*,n}(g) - \widetilde{\mathcal{R}}_{w^*,n}(g)\right| \le \frac{|n'_+/n - p'|}{\varepsilon^2}\left(1 + |n'_+/n - p'|\left(\frac{p}{n'_+/n} + \frac{1-p}{1-n'_+/n}\right)\right).
$$

By virtue of Hoeffding inequality, we obtain that, for any $\delta \in (0, 1)$, we have with probability larger than $1 - \delta$:

$$
\left|n'_+/n - p'\right| \le \sqrt{\frac{\log(2/\delta)}{2n}},
$$

so that, in particular, $\min\{n'_+/n,\ 1 - n'_+/n\} \ge \varepsilon - \sqrt{\log(2/\delta)/(2n)}$. This yields the desired result.

PROOF OF COROLLARY 1

Observe first that $\|\Phi\|_\infty \le \max(p,\ 1 - p)/\varepsilon$ and

$$
\mathcal{R}_P(\widetilde{g}_n) - \inf_{g \in \mathcal{G}} \mathcal{R}_P(g) \le 2 \sup_{g \in \mathcal{G}}\left|\widetilde{\mathcal{R}}_{\widehat{w}^*,n}(g) - \widetilde{\mathcal{R}}_{w^*,n}(g)\right| + 2 \sup_{g \in \mathcal{G}}\left|\widetilde{\mathcal{R}}_{w^*,n}(g) - \mathcal{R}_P(g)\right|.
$$

The result then directly follows from the application of Lemmas 1-2 combined with the union bound.

PROOF OF THEOREM 1

Observe first that $\|\Phi\|_\infty \le \max_k p_k/\varepsilon$ and

$$
\mathcal{R}_P(\widetilde{\theta}_n^*) - \inf_{\theta \in \Theta} \mathcal{R}_P(\theta) \le 2 \sup_{\theta \in \Theta}\left|\widetilde{\mathcal{R}}_{\widehat{w}^*,n}(\theta) - \widetilde{\mathcal{R}}_{w^*,n}(\theta)\right| + 2 \sup_{\theta \in \Theta}\left|\widetilde{\mathcal{R}}_{w^*,n}(\theta) - \mathcal{R}_P(\theta)\right|.
$$

The result then directly follows from the application of Lemmas 1-3 combined with the union bound.

**Lemma 3.** *Let $\varepsilon \in (0,\ 1/2)$. Suppose that $p'_k \in (\varepsilon,\ 1 - \varepsilon)$ for $k \in \{1,\ \ldots,\ K\}$. For any $\delta \in (0, 1)$, we have with probability larger than $1 - \delta$:*

$$
\sup_{\theta \in \Theta}\left|\widetilde{\mathcal{R}}_{\widehat{w}^*,n}(\theta) - \widetilde{\mathcal{R}}_{w^*,n}(\theta)\right| \le \frac{2L}{\varepsilon^2}\sqrt{\frac{\log(2K/\delta)}{2n}},
$$

*as soon as $n \ge 2\log(2K/\delta)/\varepsilon^2$, where $L = \sup_{(\theta,z) \in \Theta \times \mathcal{Z}} \ell(\theta, z)$.*

PROOF.

Apply the Taylor expansion

$$
\frac{1}{x} = \frac{1}{a} - \frac{x-a}{a^2} + \frac{(x-a)^2}{xa^2},
$$

so as to get for all $k \in \{1, \ldots, K\}$

$$\frac{1}{n_k'/n} = \frac{1}{p_k'} - \frac{n_k'/n - p_k'}{p_k'^2} + \frac{(n_k'/n - p_k')^2}{p_k'^2 n_k'/n}.$$

This yields the decomposition

$$\widetilde{\mathcal{R}}_{\widehat{w}^*, n}(\theta) - \widetilde{\mathcal{R}}_{w^*, n}(\theta) = \frac{1}{n} \sum_{i=1}^{n} \ell(\theta, Z_i') \sum_{k=1}^{K} \mathbb{I}\{S_i' = k\} \left( -\frac{p_k}{p_k'^2} \left( \frac{n_k'}{n} - p_k' \right) + \frac{p_k(n_k'/n - p_k')^2}{p_k'^2 n_k'/n} \right).$$

We deduce that

$$\left| \widetilde{\mathcal{R}}_{\widehat{w}^*, n}(\theta) - \widetilde{\mathcal{R}}_{w^*, n}(\theta) \right| \leq \frac{L}{\varepsilon^2} \sum_{k=1}^{K} |n_k'/n - p_k'| p_k \left( 1 + \frac{|n_k'/n - p_k'|}{n_k'/n} \right).$$

By virtue of Hoeffding inequality, we obtain that, for any $k \in \{1, \ldots, K\}$ and $\delta \in (0, 1)$, we have with probability larger than $1 - \delta$:

$$|n_k'/n - p_k'| \leq \sqrt{\frac{\log(2/\delta)}{2n}},$$

so that, by a union bound, $\max_k \{n_k'/n\} \geq \varepsilon - \sqrt{\log(2K/\delta)/(2n)}$. This yields the desired result.

PROOF OF THEOREM 2

Observe first that $\|\Phi\|_\infty \leq \max(2p, 1)/\varepsilon$ and

$$\mathcal{R}_P(\widetilde{g}_n) - \inf_{g \in \mathcal{G}} \mathcal{R}_P(g) \leq 2 \sup_{g \in \mathcal{G}} \left| \widetilde{\mathcal{R}}_{\widehat{w}^*, n}(g) - \widetilde{\mathcal{R}}_{w^*, n}(g) \right| + 2 \sup_{g \in \mathcal{G}} \left| \widetilde{\mathcal{R}}_{w^*, n}(g) - \mathcal{R}_P(g) \right|,$$

with weighted empirical risk $\widetilde{\mathcal{R}}_{w^*, n}(g)$ defined in (11). The result then directly follows from the application of Lemmas 1-4 combined with the union bound.

**Lemma 4.** *Let $\varepsilon \in (0, 1/2)$. Suppose that $q \in (\varepsilon, 1 - \varepsilon)$. For any $\delta \in (0, 1)$, we have with probability larger than $1 - \delta$:*

$$\sup_{g \in \mathcal{G}} \left| \widetilde{\mathcal{R}}_{\widehat{w}^*, n}(g) - \widetilde{\mathcal{R}}_{w^*, n}(g) \right| \leq \frac{2(2p + 1)}{\varepsilon^2} \sqrt{\frac{\log(2/\delta)}{2n}},$$

*as soon as $n \geq 2 \log(2/\delta)/\varepsilon^2$.*

PROOF. Apply twice the Taylor expansion

$$\frac{1}{x} = \frac{1}{a} - \frac{x - a}{a^2} + \frac{(x - a)^2}{x a^2},$$

so as to get

$$\frac{1}{n_+'/n} = \frac{1}{q} - \frac{n_+'/n - q}{q^2} + \frac{(n_+'/n - q)^2}{q^2 n_+'/n},$$

$$\frac{1}{n_-'/n} = \frac{1}{1 - q} - \frac{n_-'/n - 1 + q}{(1 - q)^2} + \frac{(n_-'/n - 1 + q)^2}{(1 - q)^2 n_-'/n}.$$

This yields the decomposition

$$\widetilde{\mathcal{R}}_{\widehat{w}^*, n}(g) - \widetilde{\mathcal{R}}_{w^*, n}(g) = -\frac{2p}{q^2} \left( \frac{n_+'}{n} - q \right) \frac{1}{n} \sum_{i=1}^{n} \mathbb{I}\{g(X_i') = -1, \, Y_i' = +1\}$$

$$-\frac{1}{(1 - q)^2} \left( \frac{n_-'}{n} - 1 + q \right) \frac{1}{n} \sum_{i=1}^{n} \mathbb{I}\{g(X_i') = +1, \, Y_i' = -1\} + \frac{2p(n_+'/n - q)^2}{q^2 n_+'/n} \frac{1}{n} \sum_{i=1}^{n} \mathbb{I}\{g(X_i') = -1, \, Y_i' = +1\}$$

$$+ \frac{(n_-'/n - 1 + q)^2}{(1 - q)^2 n_-'/n} \frac{1}{n} \sum_{i=1}^{n} \mathbb{I}\{g(X_i') = +1, \, Y_i' = -1\}.$$

We deduce that

$$\left|\widetilde{\mathcal{R}}_{\widehat{w^*},n}(g) - \widetilde{\mathcal{R}}_{w^*,n}(g)\right| \leq \frac{|n'_+/n - q|}{\varepsilon^2}\left(2p + 1 + |n'_+/n - q|\left(\frac{2p}{n'_+/n} + \frac{1}{1 - n'_+/n}\right)\right).$$

By virtue of Hoeffding inequality, we obtain that, for any $\delta \in (0, 1)$, we have with probability larger than $1 - \delta$:

$$\left|n'_+/n - q\right| \leq \sqrt{\frac{\log(2/\delta)}{2n}},$$

so that, in particular, $\min\{n'_+/n, \ 1 - n'_+/n\} \geq \varepsilon - \sqrt{\log(2/\delta)/(2n)}$. This yields the desired result.

ALTERNATIVE APPROACH FOR POSITIVE-UNLABELED LEARNING

**Theorem 3.** *Let $\widehat{w}^*_i = \widehat{\Phi}(X'_i, Y'_i)$ for all $i \in \{1, \ldots, n\}$ with $\widehat{\Phi}$ defined in (13). Suppose that the hypotheses of Lemma 5 are fulfilled. Let $\widetilde{g}_n$ be any minimizer of $\widetilde{\mathcal{R}}_{\widehat{w^*},n}$ over class $\mathcal{G}$. We have with probability at least $1 - \delta$:*

$$\mathcal{R}_P(\widetilde{g}_n) - \inf_{g \in \mathcal{G}} \mathcal{R}_P(g) \leq \frac{2\max(p, 1-p)}{\varepsilon}\left(\mathbb{E}[R'_n(\mathcal{G})] + \sqrt{\frac{2\log(2/\delta)}{n}}\right) + \frac{4}{\varepsilon^2}\sqrt{\frac{\log(4/\delta)}{2n}} + 4\sup_{x \in \mathcal{X}}|\widehat{\eta}(x) - \eta(x)|,$$

*as soon as $n \geq 2\log(4/\delta)/\varepsilon^2$.*

PROOF. Observe first that $\|\Phi\|_\infty \leq \max_k p_k/\varepsilon$ and

$$\mathcal{R}_P(\widetilde{\theta}^*_n) - \inf_{\theta \in \Theta} \mathcal{R}_P(\theta) \leq 2\sup_{\theta \in \Theta}\left|\widetilde{\mathcal{R}}_{\widehat{w^*},n}(\theta) - \widetilde{\mathcal{R}}_{w^*,n}(\theta)\right| + 2\sup_{\theta \in \Theta}\left|\widetilde{\mathcal{R}}_{w^*,n}(\theta) - \mathcal{R}_P(\theta)\right|.$$

The result then directly follows from the application of Lemmas 1-5 combined with the union bound.

**Lemma 5.** *Let weights $\widehat{w}^*$ be defined as in Theorem 3. Let $\varepsilon \in (0, \ 1/2)$. Suppose that $q \in (\varepsilon, \ 1 - \varepsilon)$. For any $\delta \in (0, 1)$, we have with probability larger than $1 - \delta$:*

$$\sup_{g \in \mathcal{G}}\left|\widetilde{\mathcal{R}}_{\widehat{w^*},n}(g) - \widetilde{\mathcal{R}}_{w^*,n}(g)\right| \leq \frac{2}{\varepsilon^2}\sqrt{\frac{\log(2/\delta)}{2n}} + 2\sup_{x \in \mathcal{X}}|\widehat{\eta}(x) - \eta(x)|,$$

*as soon as $n \geq 2\log(2/\delta)/\varepsilon^2$.*

PROOF.

Apply twice the Taylor expansion

$$\frac{1}{x} = \frac{1}{a} - \frac{x - a}{a^2} + \frac{(x - a)^2}{xa^2},$$

so as to get

$$\frac{1}{n'_+/n} = \frac{1}{q} - \frac{n'_+/n - q}{q^2} + \frac{(n'_+/n - q)^2}{q^2 n'_+/n},$$

$$\frac{1}{n'_-/n} = \frac{1}{1-q} - \frac{n'_-/n - 1 + q}{(1-q)^2} + \frac{(n'_-/n - 1 + q)^2}{(1-q)^2 n'_-/n}.$$

This yields the decomposition

$$\widetilde{\mathcal{R}}_{\widehat{w^*},n}(g) - \widetilde{\mathcal{R}}_{w^*,n}(g) = -\frac{p}{q^2}\left(\frac{n'_+}{n} - q\right)\frac{1}{n}\sum_{i=1}^{n}\mathbb{I}\{g(X'_i) = -1, \ Y'_i = +1\}$$

$$-\frac{1}{(1-q)^2}\left(\frac{n'_-}{n} - 1 + q\right)\frac{1}{n}\sum_{i=1}^{n}(1 - \widehat{\eta}(X'_i))\mathbb{I}\{g(X'_i) = +1, \ Y'_i = -1\} + \frac{p(n'_+/n - q)^2}{q^2 n'_+/n}\frac{1}{n}\sum_{i=1}^{n}\mathbb{I}\{g(X'_i) = -1, \ Y'_i = +1\}$$

$$+\frac{(n'_-/n - 1 + q)^2}{(1-q)^2 n'_-/n}\frac{1}{n}\sum_{i=1}^{n}(1 - \widehat{\eta}(X'_i))\mathbb{I}\{g(X'_i) = +1, \ Y'_i = -1\} + \frac{1}{(1-q)n}\sum_{i=1}^{n}(\eta(X'_i) - \widehat{\eta}(X'_i))\mathbb{I}\{g(X'_i) = +1, \ Y'_i = -1\}.$$

We deduce that

$$\left|\widetilde{\mathcal{R}}_{\widehat{w}^*,n}(g) - \widetilde{\mathcal{R}}_{w^*,n}(g)\right| \leq \frac{|n'_+/n - q|}{\varepsilon^2}\left(1 + |n'_+/n - q|\left(\frac{1}{n'_+/n} + \frac{1}{1 - n'_+/n}\right)\right) + \frac{n'_-/n}{1-q}\sup_{x\in\mathcal{X}}|\widehat{\eta}(x) - \eta(x)|.$$

By virtue of Hoeffding inequality, we obtain that, for any $\delta \in (0, 1)$, we have with probability larger than $1 - \delta$:

$$\left|n'_+/n - q\right| \leq \sqrt{\frac{\log(2/\delta)}{2n}},$$

so that, in particular, $\min\{n'_+/n, \ 1 - n'_+/n\} \geq \varepsilon - \sqrt{\log(2/\delta)/(2n)}$. Moreover, still under this event, $n'_-/n \leq 2(1 - q)$ if $n \geq \log(2/\delta)/(2\varepsilon^2)$. This yields the desired result.

## Appendix - Extension to Iterative WERM

As highlighted in Remark 2, the importance function can be expressed as a function of the ideal decision function in certain situations: Eq. (12) involves the regression function $\eta(x)$, that defines the optimal (Bayes) classifier $g^*(x) = 2\mathbb{I}\{\eta(x) \geq 1/2\} - 1$. This simple observation paves the way for a possible incremental application of the WERM approach: in the case where the solution of the WERM problem considered outputs an estimate of the optimal decision function, it can be next re-used for defining and solving a novel WERM problem. Whereas binary classification based on PU data only aims at recovering a single level set of the posterior probability $\eta(x)$, it is not the case of a more ambitious statistical learning problem, referred to as *bipartite ranking*, for which such an incremental version of WERM can be described.

**Bipartite ranking based on PU data.** In bipartite ranking, the statistical challenge consists of ranking all the instances $x \in \mathcal{X}$ through a *scoring function* $s : \mathcal{X} \to \mathbb{R}$ in the same order as the likelihood ratio $\Psi(X) = (dF_+/dF_-)(X)$, or, equivalently, as the regression function $\eta(x) = \mathbb{P}\{Y = +1 \mid X = x\}$, $x \in \mathcal{X}$: the higher the score $s(X)$, the more likely one should observe $Y = +1$. Let $\mathcal{S} = \{s : \mathcal{X} \to \mathbb{R} \text{ measurable}\}$ denotes the set of all scoring functions on the input space $\mathcal{X}$. A classical way of measuring "how much stochastically larger" a distribution $\mathcal{G}$ on $\mathbb{R}$ than another one, $\mathcal{H}$ say, consists in drawing the "probability-probability plot":

$$t \in \mathbb{R} \mapsto (1 - \mathcal{H}(t), \ 1 - \mathcal{G}(t)),$$

with the convention that possible jumps are connected by line segments (in order to guarantee the continuity of the curve). Equipped with this convention, this boils down to plot the graph of the mapping

$$\mathrm{ROC}_{\mathcal{H},\mathcal{G}} : \alpha \in (0, 1) \mapsto \mathrm{ROC}_{\mathcal{H},\mathcal{G}} = 1 - \mathcal{G} \circ \mathcal{H}^{-1}(1 - \alpha),$$

where $\Gamma^{-1}(u) = \inf\{t \in \mathbb{R} : \ \Gamma(t) \geq u\}$ denotes the pseudo-inverse of any cumulative distribution function $\Gamma(t)$ on $\mathbb{R}$. The closer to the left upper corner of the unit square $[0, 1]^2$, the larger the distribution $\mathcal{G}$ is compared to $\mathcal{H}$ in a stochastic sense. This approach is known as ROC analysis. The gold standard for evaluating the ranking performance of a scoring function $s$ is thus the ROC curve:

$$\mathrm{ROC}_s \overset{def}{=} \mathrm{ROC}_{F_{s,-}, F_{s,+}},$$

where $F_{s,+}$ and $F_{s,-}$ denote the conditional distributions of $s(X)$ given $Y = +1$ and given $Y = -1$ respectively, *i.e.* the images of class distributions $F_+$ and $F_-$ by the mapping $s(x)$. Indeed, it follows from a standard Neyman-Pearson argument that the ROC curve $\mathrm{ROC}^*$ of strictly increasing transforms of $\eta(x)$ is optimal with respect to this criterion in the sense that:

$$\forall \alpha \in (0, 1), \quad \mathrm{ROC}_s(\alpha) \leq \mathrm{ROC}^*(\alpha),$$

for any scoring function $s$. We set $\mathcal{S}^* = \{T \circ \eta : \ T : (0, 1) :\to \mathbb{R}\}$. A summary quantity of this functional criterion that is widely used in practice is the *Area Under the* ROC *Curve* (AUC in short), given by:

$$\mathrm{AUC}(s) = \int_{\alpha=0}^{1} \mathrm{ROC}_s(\alpha) \, d\alpha,$$

for $s \in \mathcal{S}$. Beyond its scalar nature, an attractive property of this criterion lies in the fact that it can be interpreted in a probabilistic manner, insofar as we have the relation: $\forall s \in \mathcal{S}$,

$$\mathrm{AUC}(s) = \mathbb{P}\{s(X) < s(X') \mid (Y, Y') = (-1, +1)\} + \frac{1}{2}\mathbb{P}\{s(X) = s(X') \mid (Y, Y') = (-1, +1)\}.$$

Denoting by $(X_i, Y_i)$, $i \in \{1, 2\}$, independent copies of the pair $(X, Y)$ and placing ourselves in the situation where $s(X)$'s distribution is continuous, as observed in Clémençon et al. (2008), we have $\text{AUC}(s) = 1 - L_P(s)/(2p(1 - p))$, where

$$L_P(s) \stackrel{def}{=} \mathbb{P}\{(s(X_1) - s(X_2))(Y_1 - Y_2)) < 0\},$$

is the *ranking risk*, the theoretical rate of discording pairs namely, that can be viewed as a pairwise classification risk. Hence, bipartite ranking can be formulated as the problem of learning a scoring function $s$ that minimizes the ranking risk

$$L_P(s) = \mathbb{E}_{P' \otimes P'} \left[ \frac{dP}{dP'}(X_1', Y_1') \frac{dP}{dP'}(X_2', Y_2') \times \mathbb{I}\{(s(X_1') - s(X_2'))(Y_1' - Y_2')) < 0\} \right].$$

Now, using Eq. (12) and the fact that $\eta = p\Psi/(1 - p + p\Psi)$, we have:

$$\frac{dP}{dP'}(x, y) = \Phi(x, y) = \frac{p}{q}\mathbb{I}\{y = +1\} + \frac{1 - \eta(x)}{1 - q}\mathbb{I}\{y = -1\} = \frac{p}{q}\mathbb{I}\{y = +1\} + \frac{1 - p}{(1 - q)(1 - p + p\Psi(x))}\mathbb{I}\{y = -1\}.$$

Therefore, it has been shown in Clémençon & Vayatis (2009) (see Corollary 5 therein) that for any $s^*$ in $\mathcal{S}^*$,

$$\frac{dF_+}{dF_-}(X) = \frac{dF_{s^*,+}}{dF_{s^*,-}}(s^*(X)) \text{ almost-surely.}$$

For any $s$ candidate, setting $\Psi_s(x) = dF_{s,+}/dF_{s,-}(s(x))$, one can define

$$\Phi_s(x, y) = \frac{p}{q}\mathbb{I}\{y = +1\} + \frac{1 - p}{(1 - q)(1 - p + p\Psi_s(s(x)))}\mathbb{I}\{y = -1\}.$$

From this formula, it is the easy to see how an incremental use of the WERM could be implemented.

- Start from an initial guess $s$ for the optimal scoring functions (*e.g.* solve the empirical ranking risk minimization problem ignoring the bias issue)
- Estimate $\Phi_s$ from the $(X_i', Y_i')$'s and the knowledge of $p$, observing that one is not confronted with the curse of dimensionality in this case
- Solve the Weighted Empirical Ranking Risk Minimization problem using the weight function

$$\widehat{\Phi}_s(x_1, y_1)\widehat{\Phi}_s(x_2, y_2),$$

   which produces a new scoring function $s$ and iterate.

Investigating the performance of such an incremental procedure will be the subject of future research.

## APPENDIX - INACCURATE PRIOR INFORMATION ABOUT THE TEST DISTRIBUTION

As noticed in section 2, it may happen that the rate of positive instances in the target population is approximately known only. Suppose that our guess for $p$ is $\widetilde{p}$ such that $|p - \widetilde{p}| \leq \zeta$, with $\zeta \in (0, 1)$. Denote by $\widetilde{P}$ the distribution over $\mathcal{X} \times \{-1, +1\}$ under which $X$ is drawn from $\widetilde{p}F_+ + (1 - \widetilde{p})F_-$ and such that $\mathbb{P}_{(X,Y) \sim \widetilde{P}}\{Y = 1 \mid X = x\} = \mathbb{P}_{(X,Y) \sim P}\{Y = 1 \mid X = x\} = \eta(x)$.

By a change of measure we have,

$$\mathbb{P}_{\widetilde{P}}(Y \neq g(X)) = \mathbb{P}_P(Y \neq g(X)) + \mathbb{E}_P\left[\left(\frac{d\widetilde{P}}{dP}(X, Y) - 1\right)\mathbb{I}\{Y \neq g(X)\}\right],$$

which allows to bound the difference of the classification risks of $g$ under $P$ and $\widetilde{P}$:

$$\left|\mathcal{R}_{\widetilde{P}}(g) - \mathcal{R}_P(g)\right| \leq \mathbb{E}_P\left[\left|\frac{d\widetilde{P}}{dP}(X, Y) - 1\right|\right] = 2|\widetilde{p} - p| \leq 2\zeta.$$

## APPENDIX - ADDITIONAL NUMERICAL EXPERIMENTS

In this Appendix, more details about the experiments carried out are provided.

IMPORTANCE OF REWEIGHTING FOR SIMPLE DISTRIBUTIONS

Introduce a random pair $(X, Y)$ in $[0, 1] \times \{-1, +1\}$ where $X \mid Y = +1$ has for probability density function (pdf) $f_+(x) = (1 + \alpha)x^\alpha, \alpha > 0$ and $X \mid Y = -1$ has for pdf $f_-(x) = (1 + \beta)(1 - x)^\beta, \beta > 0$. As in Section 2, the train and test datasets have different class probabilities $p'$ and $p$ for $Y = +1$. The loss $\ell$ is defined as $\ell(\theta, z) = \mathbb{I}\{(x - \theta)y \geq 0\}$ where $\theta > 0$ is a learnt parameter.

The true risk can be explicitely calculated. For $\theta > 0$, we have

$$R_P(\theta) = p\theta^{1+\alpha} + (1 - p)(1 - \theta)^{1+\beta},$$

and the optimal threshold $\theta_p^*$ can be found by derivating the risk $R_P(\theta)$. The derivative is zero when $\theta$ satisfies

$$p(1 + \alpha)\theta^\alpha = (1 - p)(1 + \beta)(1 - \theta)^\beta. \tag{14}$$

Solving eq. (14) is straightforward for well-chosen values of $\alpha, \beta$, which are detailed in fig. 4. The excess error $\mathcal{E}(p', p) = R_P(\theta_{p'}^*) - R_P(\theta_p^*)$ for the diagonal entries of fig. 4 are plotted in fig. 5, in the infinite sample case.

| | $(\alpha, \beta)$ | | | |
| --- | --- | --- | --- | --- |
| | $(0, 0)$ | $(1/2, 1/2)$ | $(1, 1)$ | $(2, 2)$ |
| $\theta_p^*$ | $[0, 1]$ | $\frac{(1-p)^2}{p^2+(1-p)^2}$ | $1 - p$ | $\frac{\sqrt{1-p}}{\sqrt{p}+\sqrt{1-p}}$ |

Figure 4: Optimal parameters $\theta^*$ for different values of $\alpha, \beta$.

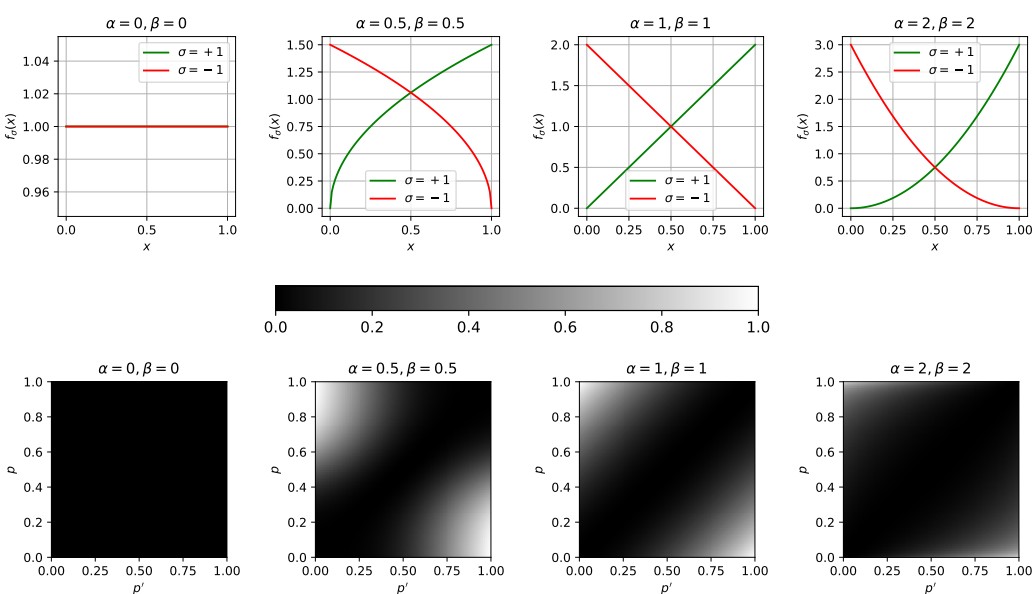

Figure 5: Pdf's and values of the excess risk $\mathcal{E}(p', p)$ for different values of $\alpha, \beta$.

The results of fig. 5 show that the optimum for the train distribution is significantly different from the optimum for the test distribution for the excess risk when the problem involves Bayes noise.

GENERALITIES ON REAL DATA EXPERIMENTS

**Strategy to induce bias in balanced datasets** In the two real data experiments described in the part concerning the MNIST experiment in the Appendix and in Section 4, the same strategy is used to induce class distribution bias or strata bias. Since both experiments involve a small test dataset, it

is kept intact, while we discard elements of the train dataset to induce bias between the train and test datasets. The bias is parameterized by a single parameter $\gamma$, such that when $\gamma$ is close to one, there is little strata or class bias, while when $\gamma$ approaches 0, bias is extreme.

The bias we induce is inspired by a power law, which is often used to model unequal distributions. Each value of a modality, *i.e.* a possible value of the stratum or class of an instance, is given by one of the values of the power law at random. Formally, the target train distribution $\{p'_k\}_{k=1}^K$ over a modality $S \in \{1, \ldots, K\}$, is defined for all $1 \le k \le K$ as

$$p'_k = \frac{\gamma^{-\frac{\lfloor K/2 \rfloor}{\sigma(k)}} p_k}{\sum_{l=1}^K \gamma^{-\frac{\lfloor K/2 \rfloor}{\sigma(k)}} p_k},$$

where $\sigma$ is a random permutation in $\{1, \ldots, K\}$.

To generate a train dataset with modality distribution $\{p'_k\}_{k=1}^K$, we sample instances from the original train data set $\mathcal{D}_n^\circ = \{(X'_i, Y'_i, S'_i)\}_{i=1}^n$, where $Y'_i$ is the class, $S'_i$ is the modality. For MNIST experiment, $S'_i = Y'_i$, while for Section 4, the value $S'_i$ is the stratum of the instance $i$. The output of the train dataset is noted $\mathcal{D}_n$, see Algorithm 1 for the detailed algorithm of the train dataset generation.

---

**Algorithm 1** Biased training dataset generation

---

**Data:** $\mathcal{D}_n^\circ = \{(X'_i, Y'_i, S'_i)\}_{i=1}^n, \{p'_k\}_{k=1}^K$
**Result:** $\mathcal{D}_n$
$D \leftarrow \emptyset$      *# Initialize the result index set.*
**for** $k = 1, \ldots, K$ **do**
  $\mathcal{I}_k \leftarrow \{i \mid 1 \le i \le n, S'_i = k\}$     *# Count the candidates for each modality.*
**end**
$m_{\text{samp}} \leftarrow \min(\#\mathcal{I}_1, \ldots, \#\mathcal{I}_K)$
**while** $m_{samp} > 0$ **do**
  $m_1, \ldots, m_K \leftarrow \mathcal{M}(m_{\text{samp}}, p_1, \ldots, p_K)$        *# $\mathcal{M}$ is the multinomial law.*
  **for** $k = 1, \ldots, K$ **do**
    $D_k \leftarrow \text{RandSet}(m_k, \mathcal{I}_k)$     *# RandSet(n, X) is a random subset of n elements of X.*
    $\mathcal{I}_k \leftarrow \mathcal{I}_k \setminus D_k$
  **end**
  $m_{\text{samp}} \leftarrow \min(\#\mathcal{I}_1, \ldots, \#\mathcal{I}_K)$
  $D \leftarrow D \cup \left( \bigcup_{k=1}^K D_k \right)$
**end**
$\mathcal{D}_n \leftarrow \{(X'_i, Y'_i, S'_i) \mid i \in D\}$
**Return** $\mathcal{D}_n$

---

**Models**   Both MNIST and ImageNet experiments compare two models: a linear model and a multilayer perceptron (MLP) with one hidden layer. Given a classification problem of input $x$ of dimension $d$ with $K$ classes, precisely with $d = 784, K = 10$ for MNIST and $d = 2048, K = 1000$ for ImageNet data, a linear model simply learns the weights matrix $W \in \mathbb{R}^{d \times K}$ and the bias vector $b \in \mathbb{R}^K$ and outputs logits $l = W^\top x + b$. On the other hand, the MLP has a hidden layer of dimension $h = \lfloor (d + K)/2 \rfloor$ and learns the weights matrices $W_1 \in \mathbb{R}^{d,h}, W_2 \in \{h, K\}$ and bias vectors $b_1 \in \mathbb{R}^h, b_2 \in \mathbb{R}^K$ and outputs logits $l = W_2^\top h(W_1^\top x + b_1) + b_2$ where $h$ is the ReLU function, i.e. $h : x \mapsto \max(x, 0)$. The number of parameters for each dataset and each model is summarized in table 1.

The weight decay or l2 penalization for the linear model and MLP model are written, respectively

$$\mathcal{P} = \frac{1}{2}\|W\| \quad \text{and} \quad \mathcal{P} = \frac{1}{2}\|W_1\| + \frac{1}{2}\|W_1\|.$$

**Cost function**   The cost function is the Softmax Cross-Entropy (SCE), which is the most used classification loss in deep learning. Specifically, given logits $l = (l_1, \ldots, l_K) \in \mathbb{R}^K$, the softmax function is $\gamma : \mathbb{R}^k \to [0, 1]^K$ with $\gamma = (\gamma_1, \ldots, \gamma_K)$ and for all $k \in \{1, \ldots, K\}$,

$$\gamma_k : l \mapsto \frac{\exp(l_k)}{\sum_{j=0}^K \exp(l_j)}.$$

|          | Model   |           |
| -------- | ------- | --------- |
| Database | Linear  | MLP       |
| MNIST    | 7,850   | 315,625   |
| ImageNet | 2,049,000 | 4,647,676 |

Table 1: Number of parameters for each model.

| Experiment | MNIST - Section 5 | ImageNet - Section 4 |
| ---------- | ----------------- | -------------------- |
| Net weights std init $\sigma_0$ | 0.01 | 0.01 |
| Weight decay $\lambda$ Unif | 0.01 | 0.002 |
| Weight decay $\lambda$ Strata | X | 0.003 |
| Weight decay $\lambda$ Class | 0.01 | 0.003 |
| Weight decay $\lambda$ Sym data | X | 0.001 |
| Learning rate $\eta$ | 0.01 | 0.001 |
| Momentum $\gamma$ | 0.9 | 0.9 |
| Batch size $B$ | 1,000 | 1,000 |
| MLP hidden layer size $h$ | 397 | 1,524 |

Figure 6: Parameters of the MNIST and ImageNet experiments - Section 5 and Section 4.

Given an instance with logits $l$ and ground truth class value $y$, the expression of the softmax cross-entropy $c(l, y)$ is

$$c(l, y) = \sum_{k=1}^{K} \mathbb{I}\{y = k\} \log\left(\gamma_k(l)\right).$$

The loss that is reweighted depending on the cases as described in Section 3 is this quantity $c(l, y)$. The loss on the test set is never reweighted, since the test set is the target distribution. The weights and bias of the model that yield the logits are tuned using backpropagation on this loss averaged on random batches of $B$ elements of the training data summed with the regularization term $\lambda \cdot \mathcal{P}$ where $\lambda$ is a hyperparameter that controls the strength of the regularization.

**Preprocessing, optimization, parameters** The images of ILSVRC were encoded using the implementation of ResNet50 provided by the library *keras*[2], see Chollet et al. (2015), by taking the flattened output of the last convolutional layer.

Optimization is performed using a momentum batch gradient descent algorithm, which updates the parameters $\theta_t$ at timestep $t$ with an update vector $v_t$ by performing the following operations:

$$v_t = \gamma v_{t-1} + \eta \nabla C(\theta_{t-1}),$$
$$\theta_t = \theta_{t-1} - v_t,$$

where $\eta$ is the learning rate and $\gamma$ is the momentum, as explained in Ruder (2016).

The parameters of the learning processes are summarized in fig. 6. The weight decay parameters $\lambda$ were cross-validated by trying values on a logarithmic scale, e.g. for ImageNet $\{10^{-4}, 10^{-3}, 10^{-2}, 10^{-1}, 1\}$ and then trying more fine-grained values between the two best results, e.g. for ImageNet $10^{-3}$ was best and $10^{-2}$ was second best so we tried $\{0.002, 0.003, 0.004, 0.005\}$. The standard deviation initialization of the weights was chosen by trial-and-error to avoid overflows. The learning rate was fixed after trying different values to have fast convergence while keeping good convergence properties.

**Stratified information for ImageNet** In this section, we detail the data preprocessing necessary to assign strata to the ILSVRC data. These were constructed using a list of 27 high-level categories found on the ImageNet website[3] copied in Figure 7. Each ILSVRC image has a ground truth low

---

[2]https://keras.io/applications/
[3] http://www.image-net.org/about-stats

| categories | # synset | # images per synset | Total # images |
|---|---|---|---|
| amphibian | 94 | 591 | 56K |
| animal | 3822 | 732 | 2799K |
| appliance | 51 | 1164 | 59K |
| bird | 856 | 949 | 812K |
| covering | 946 | 819 | 774K |
| device | 2385 | 675 | 1610K |
| fabric | 262 | 690 | 181K |
| fish | 566 | 494 | 280K |
| flower | 462 | 735 | 339K |
| food | 1495 | 670 | 1001K |
| fruit | 309 | 607 | 188K |
| fungus | 303 | 453 | 137K |
| furniture | 187 | 1043 | 195K |
| geological formation | 151 | 838 | 127K |
| invertebrate | 728 | 573 | 417K |
| mammal | 1138 | 821 | 934K |
| musical instrument | 157 | 891 | 140K |
| plant | 1666 | 600 | 999K |
| reptile | 268 | 707 | 190K |
| sport | 166 | 1207 | 200K |
| structure | 1239 | 763 | 946K |
| tool | 316 | 551 | 174K |
| tree | 993 | 568 | 564K |
| utensil | 86 | 912 | 78K |
| vegetable | 176 | 764 | 135K |
| vehicle | 481 | 778 | 374K |
| person | 2035 | 468 | 952K |

Figure 7: Original categories used to construct the strata for the experiment of Section 4.

level synset, either from in the name of the training instance, or in the validation textfile for the validation dataset, that is provided by the ImageNet website. The ImageNet API [4] provides the hierarchy of synsets in the form of *is-a* relationships, e.g. *a flamingo is a bird*. Using this information, for each synset in the validation and training database, we gathered all of its ancestors in the hierarchy that were in the table fig. 7. Most of the synsets had only one ancestor, which then accounts for one stratum. Some of the synsets had no ancestors, or even several ancestors in the table, which accounts in for extra strata, either a *no-category* stratum or a strata composed of the union of several ancestors. The final distribution of the dataset over the created strata is summarized by Figure 8. Observe the presence of a *no_strata* stratum and of unions of two high-level synsets strata, e.g. *n00015388_n01905661*. A definition provided by the API of each of the synsets involved in the strata

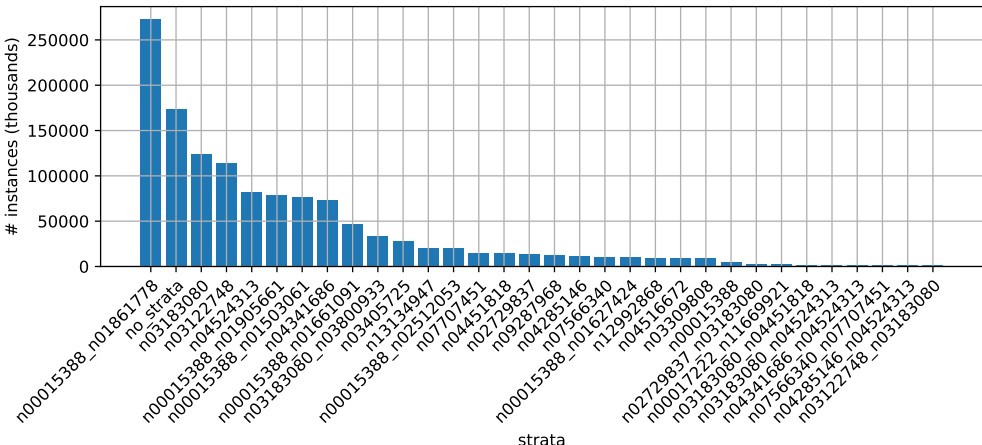

Figure 8: Distribution of the ImageNet train dataset over the created strata which definitions are given in fig. 9 .

is given in fig. 9.

### CLASSES BIAS EXPERIMENT FOR MNIST

The impact of the bias correction in the multi-class supervised learning setting described in Section 2 is illustrated on a widely used dataset for studying classification tasks: the MNIST dataset.

The MNIST dataset is composed of images $X \in \mathbb{R}^d$ of digits and labels being the value of the digits. In our experiment, we learn to predict the value of the digit so we have $K = 10$ classes corresponding to digits between 0 and 9. The dataset contains $60,000$ images for training and $10,000$ images for testing, all equally distributed within the classes. There is therefore no class bias between train and test samples in the original dataset.

Bias between classes is induced using the power law strategy described above. We deal with the classification task associated to $(X, Y)$ with a linear model or a MLP with one hidden layer that optimizes the softmax cross-entropy (SCE) using momentum gradient descent. We compare the uniform weighting of each instance in the train set (corresponding to the case where there is no reweighting described in eq. (2)) with the reweighting of each instance using the proportion of each label $Y$ for the train and test datasets as described in eq. (7). Precisions about the model and parameters are given in paragraph dealing with *Preprocessing, optimization, parameters* in the Appendix.

The optimization dynamics are summarized in fig. 10. We report the median over 100 runs of these values for the test set and a fixed random sample of the train set. For the test set, we represent 95% confidence-intervals in a lighter tone. The x-axis corresponds to the number of iterations of the learning process.

---

[4] http://image-net.org/download-API

| Strata name | | Definition | |
|---|---|---|---|
| n00015388 | n01861778 | animal, animate being, beast (...) | mammal, mammalian |
| no strata | | | |
| n03183080 | | device | |
| n03122748 | | covering | |
| n04524313 | | vehicle | |
| n00015388 | n01905661 | animal, animate being, beast (...) | invertebrate |
| n00015388 | n01503061 | animal, animate being, beast (...) | bird |
| n04341686 | | structure, construction | |
| n00015388 | n01661091 | animal, animate being, beast (...) | reptile, reptilian |
| n03183080 | n03800933 | device | musical instrument, instrument |
| n03405725 | | furniture, piece of furniture, (...) | |
| n13134947 | | fruit | |
| n00015388 | n02512053 | animal, animate being, beast (...) | fish |
| n07707451 | | vegetable, veggie, veg | |
| n04451818 | | tool | |
| n02729837 | | appliance | |
| n09287968 | | geological formation, formation | |
| n04285146 | | sports equipment | |
| n00015388 | n01627424 | animal, animate being, beast (...) | amphibian |
| n07566340 | | foodstuff, food product | |
| n12992868 | | fungus | |
| n04516672 | | utensil | |
| n03309808 | | fabric, cloth, material, textile | |
| n00015388 | | animal, animate being, beast (...) | |
| n00017222 | n11669921 | plant, flora, plant life | flower |
| n02729837 | n03183080 | appliance | device |
| n03183080 | n04451818 | device | tool |
| n03122748 | n03183080 | covering | device |
| n04285146 | n04524313 | sports equipment | vehicle |
| n04341686 | n04524313 | structure, construction | vehicle |
| n07566340 | n07707451 | foodstuff, food product | vegetable, veggie, veg |
| n03183080 | n04524313 | device | vehicle |

Figure 9: Definitions of the strata created for the experiments in Section 4, which frequencies are given in fig. 8.

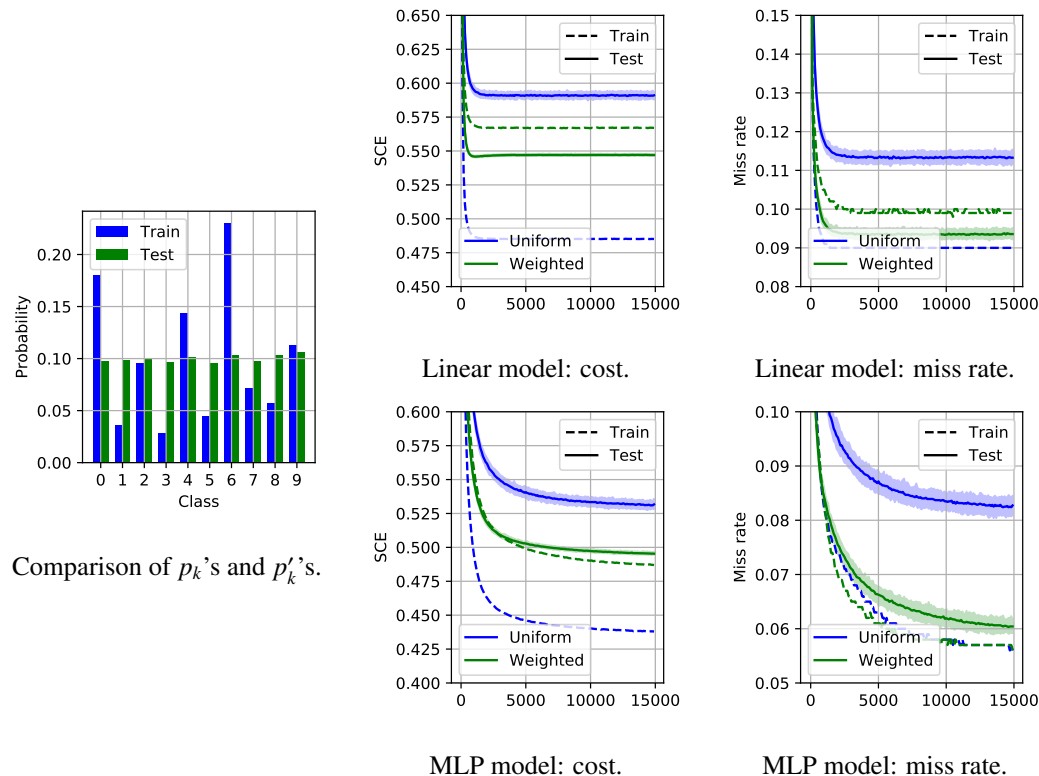

Comparison of $p_k$'s and $p'_k$'s.

Linear model: cost.

Linear model: miss rate.

MLP model: cost.

MLP model: miss rate.

Figure 10: Dynamics for the class reweighting experiment with MNIST.

For the uniform weights, we see that the misclassification rate is pretty low for the train set, but poor for the test set. By reweighting the instances, we see that we favor low error over the test set, which gives a miss probability reduced by half.

