# OpenReview forum: "Weighted Empirical Risk Minimization: Transfer Learning based on Importance Sampling"
_ICLR.cc/2020/Conference — Reject_

### Official Review · AnonReviewer3 · 2019-10-21
**Official Blind Review #3**

**Rating:** 3

**Review:**

Summary: This paper aims to show that we can estimate a density ratio for using the importance weighted ERM from the given sample and some auxiliary information on the population. Several learning bounds were proven to promote the use of importance weighted ERM.

========================================================
Clarity:
This paper is mathematically concise and understandable overall. Here, I list some comments on the clarity.

1. I found that the word auxiliary information has been used extensively from the very beginning when referring to the estimation of the density ratio. However, there is no explanation what kind of auxiliary information we need to achieve this goal until page 5, where the authors discussed about strata random variable as the additional information (if I didn't make a mistake). I believe there is a better way to introduce the intuition about what kind of auxiliary information is sufficient to make learning possible.

2. For the proposed PU-learning setting (case-control) by du Plessis et al. 2014, the assumption is the marginal density unlabeled data is identical to the test marginal density and the positive data is drawn from the class-conditional probability p(x|y=1). I am not sure if it's appropriate to discuss about it in page 2, where authors want to discuss about the situation where the train stage and test stage have different class probabilities.

3. In page 3, authors suggested that
"it is very common that the fraction of positive instances in the training dataset is significantly lower than the test set (p' < p), supposed to be known here". I have two questions about this.
3.1 Does this mean we suppose to know p', p, or both? I am aware that the appendix discussed about when p is misspecified.
3.2  I am not convinced that it is common that p' < p. It maybe nice to cite some findings or provide more explanation why it is the case.

4. In page 4, authors mentioned few shot learning problem, then describe that it is a scenario where almost no training data with positive labels is available. Is this the same problem setting as the well-known few-shot learning one? In my recognition, few-shot learning is the scenario where we want to learn from small data, e.g., p can be 0.5 but we have a very small number of data but balanced (n_pos=n_neg). Instead of few-shot, I feel it might be better to use the word like "extreme class prior or extreme class probability scenario".

5. In page 3, I'm not sure why authors suddenly focused on binary classification with varying class probabilities. A bit of introduction or motivation would be helpful. As far as I understand this is learning from class-prior shift scenario (or class-prior change), which also has been considered in the literature. Authors may consider citing some work in this line and discuss the difference in the findings of the proposed results and the existing work.

========================================================
Comments:
My impression is the novelty of this paper is modest. It is known that importance weighted ERM is unbiased and consistent to the true risk. I believe there exists theoretical analysis of learning under using WERM, especially in the situation where the weight is importance weight function is known. For Lemma 2 and Corollary 1, it is suggest that p' should not be too small but also the author suggested that p' < p. I would like to know more about the setting the authors described here, e.g., what is the example of the practical p' and p.

1. Eq. (11) is identical to the proposed unbiased risk estimator of PU-learning in du Plessis et al. (NeurIPS2014). It would be better to clarify that they are equivalent (Eq. (3) of PU-learning in du Plessis et al. (NeurIPS2014)). They also provided a generalization error bound and the analysis when p is misspecified. More theoretical analysis of this empirical risk estimator for case-control PU learning (e.g., estimation error bound) can also be found in the following paper:

Niu et al. Theoretical Comparisons of Positive-Unlabeled Learning against Positive-Negative Learning, NeurIPS2016.

2. How many trials were run in the experiments? It would be nice to see the standard error not only the mean result. It is known that importance weighting method can have high variance and it might be expected that WERM may have high variance yet have better performance. It would be helpful to explain how to read the table, e.g., what is No Bias, top-5 error. Why half of the table are in gray?

Although the paper is well-written overall. I found that it is difficult to quantify a novelty of this paper. I believe the goal, as suggested by page 2, is to "set theoretical grounds for the application of ideas behind weighted ERM". As the author suggested, this approach has been studied quite extensively both theoretically and experimentally. It would be helpful to explain what is new and the relationship of the proposed methods or bounds with the existing work to highlight the novelty of this paper.

For these reasons, I vote a weak reject for this paper.

========================================================
Potential typos:
1. There are "du Plessis et al." and "Du Plessis et al." in this paper. This indicates the same person and it should be better to use only one convention (I think du Plessis is preferable).

=========================================================
After Rebuttal:
Thanks for the reader for clarifying my several questions. I have read the rebuttal.
However, I feel that in the current form, I would like to stay with the same evaluation. The clarification of the difference between theoretical results is definitely crucial to highlight the novelty of the paper. I would like to add more comments on the PU learning part. I hope the authors find the comments useful.

1. du Plessis et al. (ICML 2015) "Convex formulation for learning from positive and unlabeled data", which was already cited in the paper, suggested that if we replace the 0-1 loss to a loss that does not have the symmetric property (e.g., logistic, squared), the form of the unbiased estimator can be different from Eq. (11) in this submitted paper (please see the paper for more details).

2. Although we obtain an unbiased risk estimator by the WERM-like method, in deep learning, minimizing such a risk may lead to overfitting, as we can see from Eq. (11) that although it is a cost-sensitive risk, it still treats all unlabeled data as negative. If we have a complex enough model, to minimize the risk, a classifier may classify all unlabeled data to negative, which undoubtedly leads to overfitting. This is discussed in Kiryo et al. (NeurIPS 2017), which also has been already cited in the submitted paper too.

Next, I would like to add more comments on the experiment.

For the experiments, I appreciate the authors' effort to do experiments on such a big dataset. In that case, it may be nice to also include an experiment on a smaller dataset, e.g., MNIST, which I believe this has already been conducted but it was in the appendix, to the main body of the paper as well to strengthen the experimental results in the paper.

I think the writing in the experiment section can be improved. For example, I don't see the first paragraph, which has lots of texts, contains much information. Also, instead of suggesting a reader to see Figure 1 for comparison, we may use more space to interpret the result. If I didn't miss it, Figure 2 and Figure 3 were never explained or referred to in the main body. In that case, we may consider removing these figures and adding the result on MNIST dataset.



**Experience Assessment:**

I have published one or two papers in this area.

**Review Assessment: Checking Correctness Of Derivations And Theory:**

I assessed the sensibility of the derivations and theory.

**Review Assessment: Checking Correctness Of Experiments:**

I carefully checked the experiments.

**Review Assessment: Thoroughness In Paper Reading:**

I read the paper thoroughly.

---

> ### Author Response · Authors · 2019-11-15
> **Response to reviewer 3**
>
> **Clarity**:
>
> 1.We tried to give intuition about the type of setting in which auxiliary information is available to reweight the empirical risk at the beginning of page 2.  Here, we provide specific examples in biometrics, that we may add in the camera-ready:
>
> Example 1: In border control with facial recognition, the countries of origin of travellers are known from their passport information and one can obtain easily the proportion of each country of origin of the travellers that pass through an airport.  Strata reweighting can be used to adapt a system for a specific location for accuracy and to correct ethnicity bias.  This side information (it is not the image data) was already used by the National Institute of Standards and Technology (NIST) to evaluate the FRVT benchmark participants, see Grother et al. 2019 [3], section 3.5 (begins at page 138).
>
> Example 2: The same type of evaluation was done on iris recognition technology, where some technologies were shown to perform differently on light-colored eyes and dark-colored eyes. Since this characteristic varies in distribution depending on geographical location, it can also be exploited in strata, see Grother et al. 2018 [4], pages 63-66. In this context, there are way fewer strata than in example 1.
>
> 3.1. In the context of the sentence, i.e. for standard binary classification, it is possible to estimate $p'$ using the dataset at hand, but $p$ is supposed to be known, which is a common assumption in PU learning.
>
> 3.2. The assumption that $p'<p$ occurs in many practical transfer learning situations. It happens when a model is trained on a global population in order to be used on only a part of it where the probability $p$ of being positive is higher. For instance in medical applications where being positive means being ill and the training dataset is composed of all patients whereas the testing dataset is only composed of patients having a specific type of symptoms which increase the risk of being ill.
>
> 4.As you pointed out, some formulations see few-shot learning as learning from a small dataset, see section 2.1 in Wang et al. (2019) [5]. However, it seems that few-shot learning covers every problem where supervised information is scarce for the task, see Definition 2.2 in Wang et al. (2019) [5], e.g.  learning to classify many new images classes (big dataset) with only one/few images per task (which makes our bound very loose).  We will make our explanation clearer in the camera-ready.
>
> 5.Binary classification with varying class probabilities is introduced as a first illustration of the general reweighting scheme with importance function $\Phi$ of Section 2, as explained in the middle of page 2.  Approaches for this problem were indeed studied under the name *class-prior change* in du Plessis et al. 2012 [1].  However, we derive a finite-time bound for Eq (7) that leverages an estimate of the train class prior through $n_+', n_-'$ which means that we deal with ratios of empirical means. To our knowledge, it constitutes original work.
>
> **Comments**:
>
> While the consistency of the Importance Weighted ERM is known, the derivation of learning bounds was tackled in Cortes et al. 2010 in the case where the whole importance function is known. This setting is not practical, since knowing the importance function requires knowing the distribution of the data. We show that many scenarios in practical situations and in the literature can be seen as WERM, and they all require information on the relation between the test and train distribution.
>
> There is a difference between p' too small, i.e. `p'<<p` and `p'<p`.
>
> 1.We agree that Eq (11) is Eq (3) in du Plessis et al. (2014) and the paper is cited in the section. We will refer explicitly to Eq (3) of du Plessis et al.  (2014) in the camera-ready, before we introduce Eq (11) in our paper. Unlike our analysis, the derivations of du Plessis et al. (2014) and Niu et al. (2016) [2] focus on a specific type of functions and assume a fixed number of positive and unlabeled points.  We will compare more extensively our results to [2] in the camera-ready.
>
> 2.While we are aware of the variance of the Importance Sampling procedure, the experiments are performed on the ImageNet dataset, which contains 1.3 million images spread out over 1.000 classes. Hence, it would be computationally intensive to compute sensible standard errors for each setting.  The settings in the Table of Figure 1 are described at the end of Section 4 (after Figure 3), but we will make it clearer by explicitly referring to their names in this paragraph.
>
> Since ImageNet is a balanced dataset (does not contain stratum shift), we generated stratum-shift artifically by removing instances (modifying academic datasets is common practice, see for example https://arxiv.org/abs/1803.09797 ) with strata based on the WordNet structure. The greyed out lines are runs with the full data, which are not attainable in our stratum-shift scenario but provided as reference.

---

### Official Review · AnonReviewer2 · 2019-10-23
**Official Blind Review #2**

**Rating:** 3

**Review:**

This paper targets the transfer learning problem. It wants to construct the unbiased estimator of the true risk for the target domain based on data from the source domain. To have the unbiased estimator, samples in the source domain are weighted based on some auxiliary information of both the source domain data distribution and the target domain data distribution. Especially, similar to previous works, the paper first assumes P(Y=1) is known for the target domain, and give a generalization bound for learning on the target domain. Then they consider two more concrete problems, one is learning with stratified information when the conditional probability given the stratified information of the source domain is equal to that of the target domain. Then the paper considers PU learning. Generalization bounds are also given for these two problems. Finally, the paper shows some empirical results showing the reweighting effect of its proposal.

The paper is a theoretical study of transfer learning, and a generalization of other learning problems including transfer learning, learning from stratified data and PU learning. It assumes that when some auxiliary information is known, generalization bound can be given by only minimizing a reweighted loss of the biased source domain data. However, the auxiliary information proposed in this paper is difficult to be got. Thus, the practical use of this paper may be limited. The paper also lacks discussion with related theoretical work (such as generalization bound of PU learning). Due to these reasons, I rate a weak reject for the paper.

In Sec. 2, to have an unbiased risk estimator as well as a generalization bound, the prior probability P(Y=1) should be known. However, the paper fails to provide any practical way to estimate this value. Although in the auxiliary part, some results when such a value cannot be accurately estimated are given, estimation methods are also required for the method to be practical. Moreover, such as result is already studied in Sugiyama et al. (2008). Thus, the novelty of this part is limited.

In Sec. 3.1, the paper focuses on the learning from stratified data problem, when some stratified information s for the data are given. The paper further assumes P(x|S=k) = P(x’|S’=k). First, in a general learning problem, no matter transfer learning or not, only information of x and y is available. To justify that the information s is available, some real applications should be given as motivations. Moreover, the assumption on the stratified data, i.e. P(x|S=k) = P(x’|S’=k) and P(S=k) \neq P(S’=k) should also be justified.

In Sec. 3.2, the generalization bound of PU learning is also studied before in for example [Niu et al., Theoretical Comparisons of Positive-Unlabeled Learning against Positive-Negative Learning. NIPS 2016]. Discussion on the relationship between these theoretical results should be given. Also, in the experimental part, there is no empirical results comparing the proposed method with the existing PU learning methods. Since one of the main contributions of this paper is on PU learning, empirical studies should also be provided to show the superior of the proposed method.

----------------------------
The rebuttal is subjective (without enough support but expressions such as "we believe", "there is no point") and fails to address my concern. I will not raise my score.

**Experience Assessment:**

I have published one or two papers in this area.

**Review Assessment: Checking Correctness Of Derivations And Theory:**

I assessed the sensibility of the derivations and theory.

**Review Assessment: Checking Correctness Of Experiments:**

I assessed the sensibility of the experiments.

**Review Assessment: Thoroughness In Paper Reading:**

I read the paper at least twice and used my best judgement in assessing the paper.

---

> ### Author Response · Authors · 2019-11-15
> **Response to reviewer 2**
>
> Since many points are common with reviewer 3 (R.3), we refer to our answers to their review here.
>
> In many practical cases, the proportion of positive instances or the strata probabilities are known, as seen in **Clarity**-1 of R.3. Hence, we believe that our results are practical, even though they do not correspond to the setting in which one has no idea of the range of the class prior. The work of Sugiyama et al. (2008) provides a practical way (the Kullback-Leibler Importance Estimation Procedure - KLIEP) of learning with importance reweighting, but is limited to a specific type of functions and does not derives any theoretical guarantees.
>
> The stratum-shift case covers all the cases where the train and test distributions are both mixtures with the same components but different probability weights (stratum S = mixture component to which an observation X belongs).  See the answer in **Clarity**-1. of R.3 for specific examples. For the discussion with related work, see **Comments**-1 of R.3.
>
> The paper shows that PU learning can be seen as a specific case of WERM, and derives guarantees for PU learning. There is no point in showing that the PU learning formulation in Eq (11) (which is also the formulation of du Plessis et al. (2014) Eq (3), see R.3) performs better than other approaches for PU learning. The iterative WERM procedure in the appendix will be studied experimentally in future work.

---

### Official Review · AnonReviewer1 · 2019-10-23
**Official Blind Review #1**

**Rating:** 3

**Review:**

The authors consider the problem of a mismatch between the distribution of the training observations and the test distribution (a transfer learning setup). The paper seems technically sound but it is not easy to read. Even Section 2 it is difficult to read.

-  Main drawback: Please define the weights w_i of Eq. (5) in Section 2.

- I have a question: is it possible to extend your work considering Multiple Importance Sampling and Generalized  Multiple Importance Sampling  schemes? please discuss.




**Experience Assessment:**

I have published one or two papers in this area.

**Review Assessment: Checking Correctness Of Derivations And Theory:**

I did not assess the derivations or theory.

**Review Assessment: Checking Correctness Of Experiments:**

I did not assess the experiments.

**Review Assessment: Thoroughness In Paper Reading:**

I made a quick assessment of this paper.

---

> ### Author Response · Authors · 2019-11-15
> **Response to reviewer 1**
>
> You may find the definition of the weights w_i of Eq (5) right under Eq (3).
>
> Multiple Importance Sampling (MIS) uses several proposal functions to sample points that follow a target distribution. In the context of our work, the proposal function is the training dataset. I would interpret generalizing our work to multiple importance sampling to involve several training datasets, with different distributions.
>
> One can straightforwardly generalize our analysis to this case, and leverage different sampling probabilities between the datasets reduce the magnitude of the impact of $\|\Phi\|$ in the bound of Lemma 1.

---

### Author Response · Authors · 2019-11-15
**References and additional information for all reviewers**

We thank the reviewer for their helpful comments and remarks. Below we respond to specific points. The following articles were mentioned in the rebuttal and are not already in the paper's reference section. They will be added in the bibliography for the camera-ready paper. Typos were taken into account and corrected.

[1] Semi-Supervised Learning of Class Balance under Class-Prior Change by Distribution Matching,
     du Plessis et Sugiyama, 2012.

[2] Theoretical Comparisons of Positive-Unlabeled Learning against Positive-Negative Learning,
      Niu et al., 2016.

[3] NIST FRVT 1:1 Verification report,
     Grother et al., 2019.

[4] IREX IX Part One Performance of Iris Recognition Algorithms,
      Grother et al., 2018.

[5] Generalizing from a Few Examples: A Survey on Few-Shot Learning,
      Wang et al., 2019.

---

### Decision · Program_Chairs · 2019-12-19

**Decision:**

Reject

**Comment:**

This paper aims to address transfer learning by importance weighted ERM that estimates a density ratio from the given sample and some auxiliary information on the population. Several learning bounds were proven to promote the use of importance weighted ERM.

Reviewers and AC feel that the novelty of this paper is modest given the rich relevant literature and the practical use of this paper may be limited. The discussion with related theoretical work such as generalization bound of PU learning can be expanded significantly. The presentation can be largely improved, especially in the experiment part. The rebuttal is somewhat subjective and unconvincing to address the concerns.

Hence I recommend rejection.